# Induction Heads Interpolate N-Grams

**Francesco D'Angelo** [* 1]  **Oğuz Kaan Yüksel** [* 1]  **Swathi Shree Narashiman** [† 2]  **Nicolas Flammarion** [1]

## Abstract

Induction heads are attention circuits believed to underlie in-context learning in transformers, yet a precise characterization of the estimators they implement remains elusive. We study transformers trained on order-$k$ Markov chains and identify two complementary smoothing mechanisms. First, at finite attention-weight scale, the circuit implements a soft context-matching estimator: it aggregates contributions from exact and partial context matches, weighted exponentially by their overlap, and induces a data-dependent interpolation across context orders analogous to Jelinek–Mercer smoothing. Second, a beginning-of-sequence (BOS) token induces additive pseudo-counts, recovering Dirichlet-style smoothing. We construct a disentangled transformer implementing both mechanisms and show that trained transformers recover the predicted attention patterns. Across settings where pseudo-count smoothing is optimal or lower-order contexts provide structured evidence, trained transformers match or outperform classical count-based baselines. Our results bridge mechanistic interpretability of induction heads with classical statistical smoothing, revealing that transformers learn to regularize in-context estimation rather than simply count.

## 1. Introduction

A striking capability of large language models is in-context learning (ICL): adapting to new tasks from examples in the prompt, without any parameter updates (Brown et al., 2020; Min et al., 2022; Bubeck et al., 2023). A growing body of work explains ICL by identifying *circuits* that transformers implement in their forward pass to solve in-context tasks. Several such circuits realize principled statistical procedures:

gradient-based optimization and least-squares for regression and autoregressive systems (Garg et al., 2022; Akyürek et al., 2022; Von Oswald et al., 2023a; Zhang et al., 2024a; Sander et al., 2024; Vladarean et al., 2026), mirror descent for latent-mixture inference (D'Angelo & Flammarion, 2026), implicit Bayesian inference (Xie et al., 2022; Zhang et al., 2024b), and algorithm selection (Bai et al., 2023). For language models, the canonical circuit is the *induction head* (Elhage et al., 2021; Olsson et al., 2022): two attention layers that search the context for a matching pattern and copy the token that followed it, implementing in-context associative recall.

In controlled sequential settings such as Markov chains and $n$-grams (Bietti et al., 2023; Edelman et al., 2024; Nichani et al., 2024; Chen et al., 2024; Ekbote et al., 2026), prior analyses characterize induction-head circuits in the hard-attention limit, where attention selects exact matches and the resulting predictor reduces to maximum-likelihood (ML) counting. This counting view is mechanistically appealing but statistically incomplete: for high-order dependencies, exact $k$-gram matches are too rare to support reliable prediction, and the ML estimator assigns zero mass to every unseen continuation. Classical language models address this problem through smoothing and backoff; redistributing probability mass to unseen events and interpolating estimates across shorter context lengths (MacKay & Peto, 1995; Liu et al., 2024). The induction-head literature for transformers, by contrast, stops at unregularized counting. This raises the following question:

*Do induction heads implement a richer class of estimators beyond strict counting?*

We study this in order-$k$ Markov prediction: a setting simple enough for exact analysis yet rich enough to capture the finite-sample tradeoff faced by any context-based predictor. Each sequence is generated by a latent transition rule, and the model must predict the next token from the history.

**Contributions.** We show that the induction-heads, analyzed beyond its hard-attention limit, implement analogues of smoothing procedures developed for classical $n$-gram language models. ML $k$-gram counting is one limiting case of this circuit, recovered when the attention weights are large; at finite attention-weight scale, together with structural tokens such as BOS, the same circuit expresses a richer family of smoothed estimators.

---

[*]Equal contribution [†]Work done during an internship at the TML Lab, EPFL. [1]TML Lab, EPFL, Switzerland [2]Department of Electrical Engineering, IIT Madras, Chennai, India. Correspondence to: Francesco D'Angelo <francesco.dangelo@epfl.ch>.

*Proceedings of the 43$^{rd}$ International Conference on Machine Learning*, Seoul, South Korea. PMLR 306, 2026. Copyright 2026 by the author(s).

- **Soft context matching.** We give a constructive proof that a two-layer induction-head circuit implements a *soft context-matching estimator*. Rather than relying only on exact order-$k$ matches, it scores every candidate context in the history based on the positions where it matches the current context, allowing both contiguous and non-contiguous matches, and aggregates their next-token predictions with weights that depend exponentially on this overlap. The scale of the attention weights controls this interpolation: large weights recover hard $k$-gram counting, while finite weights spread mass toward lower-order and context-independent estimates, inducing a data-dependent interpolation over context orders analogous to Jelinek–Mercer smoothing.

- **BOS as additive pseudo-counts.** We show that a beginning-of-sequence (BOS) token, by acting as a sequence-independent attention target, allows the model to add constant pseudo-counts to the context-specific transition counts, recovering add-$\alpha$-type smoothing. This gives a circuit-level account of how a common architectural convention can implement prior-like regularization.

- **Empirical validation.** We confirm the theory on trained transformers, including fully standard ones. Trained models exploit the smoothing mechanisms available to them: without BOS, they rely on soft context matching to interpolate across context orders; with BOS, they use pseudo-counts, matching the add-$\alpha$ Bayes-optimal predictor under independent Dirichlet priors and combining it with soft context matching interpolation under hierarchical priors.

## 2. Preliminaries

We study token sequences $\boldsymbol{x} = (x_1, \ldots, x_T)$ over a finite vocabulary $V$ of size $|V|$. We identify each token $x_i \in V$ with its one-hot vector in $\{0, 1\}^{|V|}$ when needed. For a sequence $\boldsymbol{x}$, we write $\boldsymbol{x}_a^b := (x_a, x_{a+1}, \ldots, x_b)$ to denote the subsequence from position $a$ to $b$ (inclusive) and also let $u_t := (x_{t-k+1}, x_{t-k+2}, \ldots, x_t) \in V^k$ to be the length-$k$ context at time step $t$ where $k$ is a fixed integer. We denote the set $\{a, a+1, \ldots, b\}$ by $[a, b]$ for any integers $a < b$ and write $[a] := \{1, 2, \ldots, a\}$.

### 2.1. Disentangled Transformer Models

The *disentangled transformer* enhances interpretability by removing MLPs and replacing additive residual connections with concatenation. This creates an explicit residual stream that preserves the history of computations. The input token $x_i$ is fed to the model by its one-hot vector $x_i \in \{0, 1\}^{|V|}$. The model relies on the following simplified attention mechanism. For layer $l$ and head $h$, we use a single matrix

$\boldsymbol{W}_A^{(l,h)} \in \mathbb{R}^{d_{l-1} \times d_{l-1}}$. The attention scores are defined as:

$$e_{ij}^{(l,h)} = (\boldsymbol{h}_i^{(l-1)})^\top \boldsymbol{W}_A^{(l,h)} \boldsymbol{h}_j^{(l-1)} + \text{PE}_{ij}^{(l,h)}, \quad (1)$$

where $\text{PE}_{ij}^{(l,h)}$ depends on the positional encoding scheme. The layer update is defined by concatenating the head outputs with the input:

$$\hat{\boldsymbol{h}}_i^{(l,h)} = \sum_{j=1}^{T} \mathcal{A}_{ij}^{(l,h)} \boldsymbol{v}_{ij}^{(l,h)}, \quad (2)$$

$$\boldsymbol{H}^{(l)} = \text{Concat}\left(\boldsymbol{H}^{(l-1)}, \hat{\boldsymbol{H}}^{(l,1)}, \ldots, \hat{\boldsymbol{H}}^{(l,H_l)}\right). \quad (3)$$

Here, $\boldsymbol{v}_{ij}^{(l,h)}$ represents the value vector, which is set to $\boldsymbol{h}_j^{(l-1)}$. The dimensionality of the representation grows as $d_l = d_{l-1}(1 + H_l)$ where $H_l$ is the number of heads at layer $l$. After $L$ layers, a final linear layer $\boldsymbol{W}_O \in \mathbb{R}^{|V| \times d_L}$ maps the final representation $\boldsymbol{H}^{(L)}$ to output predictions. The attention weights $\mathcal{A}_{ij}^{(l,h)}$ are computed via a causally masked softmax, $\mathcal{A}_{ij}^{(l,h)} = \left[\text{softmax}\left(\boldsymbol{e}_i^{(l,h)} + \boldsymbol{m}_i\right)\right]_j$, where $\boldsymbol{m}_i \in \{0, -\infty\}^T$ is the causal mask $(\boldsymbol{m}_i)_j = 0$ for $j \leq i$ and $(\boldsymbol{m}_i)_j = -\infty$ otherwise.

**Relative Positional Encoding (RPE).** This method encodes the distance $i - j$ between tokens. With causal masking ($i \geq j$), we only need relative positions in $\{0, 1, \ldots, T-1\}$. We introduce a learnable lookup vector $\boldsymbol{R}_A^{(l,h)} \in \mathbb{R}^T$ (one scalar per relative position). We retrieve the scalar $r_{i-j}^{(l,h)}$ corresponding to index $i - j$. The positional term in the attention is:

$$\text{PE}_{ij}^{(l,h)} = r_{i-j}^{(l,h)}. \quad (4)$$

**BOS Token.** We consider settings both with and without a prepended Beginning of Sequence (BOS) token. The primary reason for using a BOS token is to provide a dedicated "sink" for attention heads, ensuring they have a valid, neutral state to attend to when no other relevant context is available. We allow the BOS token to have a fixed non-one-hot representation.

### 2.2. In-Context Learning of Markov Chains

We study the capabilities of Transformers to perform in-context learning (ICL) on sequences generated by *order-$k$* Markov chains. Each ICL task corresponds to a latent transition rule $\boldsymbol{\pi}$ drawn from a prior; conditional on $\boldsymbol{\pi}$, a sequence is generated by the induced Markov process.

**Generative Process.** Fix an order $k \geq 1$ and a vocabulary $V$. A latent task is represented by a collection of conditional distributions $\boldsymbol{\pi} = \{\boldsymbol{\pi}_u\}_{u \in V^k}$, where each $\boldsymbol{\pi}_u \in \Delta^{|V|-1}$ specifies the distribution of the next token given the length-$k$ context $u \in V^k$. At time step $t$, the context is $u_{t-1} = (x_{t-k}, \ldots, x_{t-1})$; i.e.,

$$\boldsymbol{\pi}_{u_{t-1}}(m) = \mathbb{P}(x_t = m \mid u_{t-1}), \quad m \in V.$$

The generation of a sequence $x$ proceeds as follows:

1. **Sample Task:** Sample $\pi$ from a prior distribution (specified below).
2. **Sample Sequence:** Sample an initialization $x_{1:k} \sim \text{Unif}(V)^k$. For $t = k + 1, \ldots, T$, generate tokens according to the order-$k$ Markov property:

$$x_t \mid x_{t-k:t-1} \sim \text{Categorical}(\pi_{u_{t-1}}) ,$$

We write $x \sim \pi$ for brevity.

**Task prior over transitions.** We consider two prior distributions over $\pi$.

**(a) Independent Dirichlet prior.** For each context $u \in V^k$, we sample transitions independently as

$$\pi_u \sim \text{Dirichlet}(\alpha) \quad \forall u \in V^k ,$$

where $\alpha \in \mathbb{R}_+^{|V|}$ controls sparsity/uniformity (e.g., $\alpha = \mathbf{1}$ yields a uniform prior over the simplex).

**(b) Hierarchical Dirichlet prior.** To induce smoothing across context lengths, we define a hierarchy over suffixes. First, sample a base distribution

$$\pi_{()} \sim \text{Dirichlet}(\eta_0 \cdot \mathbf{1}) ,$$

and then, for each level $\ell \in [1, k]$ and each context $c = (c_1, \ldots, c_\ell) \in V^\ell$, sample

$$\pi_{(c_1,\ldots,c_\ell)} \sim \text{Dirichlet}\big(\eta_\ell \cdot \pi_{(c_2,\ldots,c_\ell)}\big) ,$$

so that higher-order transitions are centered on their length-$(\ell - 1)$ suffix. The parameters $\eta_1, \ldots, \eta_k > 0$ control the strength of coupling: large $\eta_\ell$ concentrates $\pi_{(c_1,\ldots,c_\ell)}$ around its parent, while small $\eta_\ell$ allows greater deviation. By construction, all length-$\ell$ contexts sharing their shorter suffix are drawn around a common parent, so their transition rows are correlated. Thus, shorter suffixes provide evidence about longer contexts that share them. A predictor can exploit this through smoothing or back-off: when the full context is too rare to estimate reliably, it falls back to more frequent shorter suffixes.

**The In-Context Learning Task.** The objective of the model $f$ is to minimize the prediction error for the token $x_{T+1}$ given the context $x_1^T$, marginalized over the prior distribution of latent tasks $\pi$. Formally, the optimization problem is:

$$\inf_{f \in \mathcal{F}} \mathbb{E}_\pi \mathbb{E}_{x \sim \pi} \mathbb{E}_{x_{T+1} \sim \pi_{u_T}} \left[ -\log f(x_{T+1} \mid x_1^T) \right] . \quad (5)$$

We emphasize that we optimize only the *last-token* negative log-likelihood at $t = T + 1$ (instead of the sum over $t$), isolating the model's ability to use the preceding context $x_1^T$ for in-context inference. To minimize this loss, the model must predict $x_{T+1}$ from the observed history $x_1^T$ alone.

**Bayes-Optimal Predictor.** We consider the Bayes-optimal predictor under each of our two task priors.

**(a) Independent Dirichlet prior.** Due to the conjugacy of the Dirichlet prior with the Categorical likelihood, the Bayes-optimal solution to Equation (5) is analytically tractable in this case. For any context $u \in V^k$ and symbol $m \in V$, define the (prefix) transition count

$$N_u^{(t)}(m) := \#\Big\{ s \in [k+1, t] : u_{s-1} = u,\ x_s = m \Big\} \quad (6)$$

and $N_u^{(t)} := \sum_{m \in V} N_u^{(t)}(m)$. The posterior distribution of the row $\pi_u$ at time $t$ is

$$\mathbb{P}(\pi_u \mid x_1^t, \alpha) = \text{Dirichlet}(\alpha + N_u^{(t)}) ,$$

where $N_u^{(t)} = (N_u^{(t)}(m))_{m \in V} \in \mathbb{N}_+^{|V|}$. Thus the optimal predictive distribution for the next token $x_{t+1}$, given the current context $u_t$ is the posterior mean

$$\mathbb{E}[\pi_{u_t}(m) \mid x_1^t] = \frac{\alpha_m + N_{u_t}^{(t)}(m)}{\sum_{j \in V} (\alpha_j + N_{u_t}^{(t)}(j))} . \quad (7)$$

This implies that the optimal in-context learner effectively implements a count-based estimator of the order-$k$ transition rule with add-$\alpha$ smoothing, reducing to Laplace smoothing when $\alpha_m = 1$ for all $m \in V$.

**(b) Hierarchical Dirichlet prior.** Under the hierarchical prior, the transition distributions across contexts are *coupled* through shared parent distributions. Consequently, the Bayes-optimal predictive distribution still takes the form

$$\mathbb{P}(x_{t+1} = m \mid x_1^t) = \mathbb{E}\big[\pi_{u_t}(m) \mid x_1^t\big] ,$$

but this posterior expectation no longer admits a simple closed-form expression analogous to Equation (7) (see, e.g., MacKay & Peto, 1995).

## 3. Induction Heads as Soft Context Matchers

In this section, we characterize some estimators of interest that a two-layer disentangled transformer can implement for next-token prediction tasks on order-$k$ Markov chains.

### 3.1. Main Result

We set $k' = k + 2$ when the BOS token is included, and $k' = k + 1$ otherwise. Note that $k'$ is the smallest index with access to a full-context. To measure similarity between the query context $u_t$ and candidate contexts $u_s$ for $s \in [k', t]$, we introduce the *match mask*.

**Definition 3.1** (Mask-Conditioned Counts). For any $s \in [k', t]$, the *match mask* between $u_t$ and $u_s$ is defined as

$$M_s^{(t)} := \big\{ r \in [k] : x_{t-r+1} = x_{s-r} \big\} \subseteq [k] .$$

The cardinality $|M_s^{(t)}|$ counts the number of positions where the two contexts agree. For a fixed query position $t$ and any $M \subseteq [k]$, define the *mask* counts $N_M^{(t)}$ and the *mask-conditioned transition* counts $N_M^{(t)}(m)$:

$$N_M^{(t)} := \#\big\{ s \in [k',t] : M_s^{(t)} = M \big\},$$
$$N_M^{(t)}(m) := \#\big\{ s \in [k',t] : M_s^{(t)} = M,\ x_s = m \big\}.$$

Note that $N_M^{(t)} = \sum_{m \in V} N_M^{(t)}(m)$. The special case $M = [k]$ corresponds to *exact* context matches, and $N_{[k]}^{(t)}(m)$ coincides with the standard transition count $N_{u_t}^{(t)}(m)$ defined in Equation (6). We now state our main theoretical result, which constructs a two-layer transformer that realizes an interpolated estimator over masks counts.

**Proposition 3.1** (Transformer Estimator for Order-$k$ Markov Chains)**.** *There exists a two-layer disentangled transformer $\mathcal{T}$ with RPE using $k$ attention heads in the first layer and a single attention head in the second layer, such that for any input sequence $\boldsymbol{x} \in V^T$, the model is a probability distribution $\mathcal{T}(\boldsymbol{x}) \in \Delta^{|V|-1}$ over $V$ given by*

$$\mathcal{T}(\boldsymbol{x})(m) = \frac{e^\kappa/|V| + \sum\limits_{M \subseteq [k]} e^{|\boldsymbol{\beta}|_M} N_M^{(T)}(m)}{e^\kappa + \sum\limits_{M \subseteq [k]} e^{|\boldsymbol{\beta}|_M} N_M^{(T)}}, \quad (8)$$

*where $|\boldsymbol{\beta}|_M = \sum_{i \in M} \beta_i$, $\boldsymbol{\beta} = (\beta_1, \ldots, \beta_k) \in \mathbb{R}^k$ and $\kappa \in \mathbb{R}$ are free parameters, and $\kappa \neq -\infty$ only when BOS token is prepended.*

Two features of the estimator in Equation (8) are worth noting. First, prepending a BOS token enables an additive constant (pseudo-count) term $e^\kappa/|V|$, yielding an add-$\boldsymbol{\alpha}$-type smoothing of the empirical counts. Second, for finite attention-weight parameters $\boldsymbol{\beta}$, the factors $e^{|\boldsymbol{\beta}|_M}$ induce an interpolation across context orders, producing a Jelinek–Mercer-style estimator. Together, these provide two complementary knobs, pseudo-count smoothing and order interpolation. Section 4 focuses on these two mechanisms, and relate them to classical $n$-gram smoothing techniques.

### 3.2. Proof of Proposition 3.1

The proof is constructive: we explicitly specify all the weights of a two-layer disentangled transformer $\mathcal{T}$ that computes Equation (8). We follow the notation from Section 2.1.

### 3.2.1. LAYER 1: COPYING HEADS

The first layer uses $k$ attention heads, each implementing a "copy" operation that retrieves a specific token from the context. For head $h \in [k]$, we design the attention mechanism to copy the token at relative position $-h$. **Positional weights.** We set the content-based attention matrices

to zero, $\boldsymbol{W}_A^{(1,h)} = \boldsymbol{0}_{|V| \times |V|}$. The attention RPE vector $\boldsymbol{R}_A^{(1,h)} \in \mathbb{R}^T$ is then set to implement hard attention to relative position $h$:

$$(\boldsymbol{R}_A^{(1,h)})_{i-j} = \begin{cases} +\delta^{(1)} & \text{if } i - j = h, \\ 0 & \text{otherwise}, \end{cases}$$

where $\delta^{(1)} > 0$ is a large constant. This structure ensures that, regardless of the token content, the attention focuses on the position at lag $h$. **Attention scores and output.** The attention score from position $i$ to position $j$ for head $h$ is

$$e_{ij}^{(1,h)} = \begin{cases} +\delta^{(1)} & \text{if } i - j = h, \\ 0 & \text{otherwise}. \end{cases}$$

In the limit $\delta^{(1)} \to \infty$, the softmax yields hard attention: $\mathcal{A}_{ij}^{(1,h)} = \mathbb{I}\{j = i - h\}$ for any $i > h$. The output of head $h$ at position $i > h$ is: $\hat{\boldsymbol{h}}_i^{(1,h)} = \sum_{j=1}^{i} \mathcal{A}_{ij}^{(1,h)} \boldsymbol{h}_j^{(0)} = x_{i-h}$, whereas for position $i = 0$, we have $\hat{\boldsymbol{h}}_i^{(1,h)} = x_1$.

Using the concatenation update rule, the representation after layer 1 is for $i > k$: $\boldsymbol{h}_i^{(1)} = (x_i, x_{i-1}, \ldots, x_{i-k}) \in \{0,1\}^{(k+1)|V|}$. This representation encodes both the current token $x_i$ and the full length-$k$ context $(x_{i-1}, \ldots, x_{i-k})$. The residual stream structure is:

$$\boldsymbol{h}_i^{(1)} = \begin{pmatrix} x_i \\ x_{i-1} \\ x_{i-2} \\ \vdots \\ x_{i-k} \end{pmatrix} \begin{matrix} \leftarrow \text{current token (block 0)} \\ \leftarrow \text{lag 1} \\ \leftarrow \text{lag 2} \\ \\ \leftarrow \text{lag } k \end{matrix} \quad (9)$$

### 3.2.2. LAYER 2: CONTEXT MATCHING

The second layer uses a single attention head that compares the query context to all candidate contexts and aggregates their successor tokens. We set the attention RPE

$$\boldsymbol{R}_A^{(2)} = \begin{cases} \begin{matrix} {\scriptstyle 0 \,\cdots\, T-k' \;\; T-k'+1 \;\cdots\; T-2 \;\; T-1} \\ \begin{pmatrix} 0 & \cdots & 0 & -\delta_1^{(2)} & \cdots & -\delta_1^{(2)} & \delta_2^{(2)} \end{pmatrix} \end{matrix} & \text{w BOS}, \\[1em] \begin{matrix} {\scriptstyle 0 \,\cdots\, T-k' \;\; T-k'+1 \;\cdots\; T-1} \\ \begin{pmatrix} 0 & \cdots & 0 & -\delta_1^{(2)} & \cdots & -\delta_1^{(2)} \end{pmatrix} \end{matrix} & \text{w/o BOS}, \end{cases}$$

where $\delta_1^{(2)}, \delta_2^{(2)} \in \mathbb{R}$ are constants. We design the attention matrix $\boldsymbol{W}_A^{(2)} \in \mathbb{R}^{(k+1)|V| \times (k+1)|V|}$ to compute a shift-aligned inner product between the query and key contexts, matching each query context token with the corresponding predecessor token of the key. Partition $\boldsymbol{h}_i^{(1)}$ into $(k+1)$ blocks of size $|V|$, indexed by $r \in \{0, 1, \ldots, k\}$, where block $r$ contains $x_{i-r}$. We define:

$$\boldsymbol{W}_A^{(2)} = (\boldsymbol{S} \otimes \boldsymbol{I}_{|V|}),$$

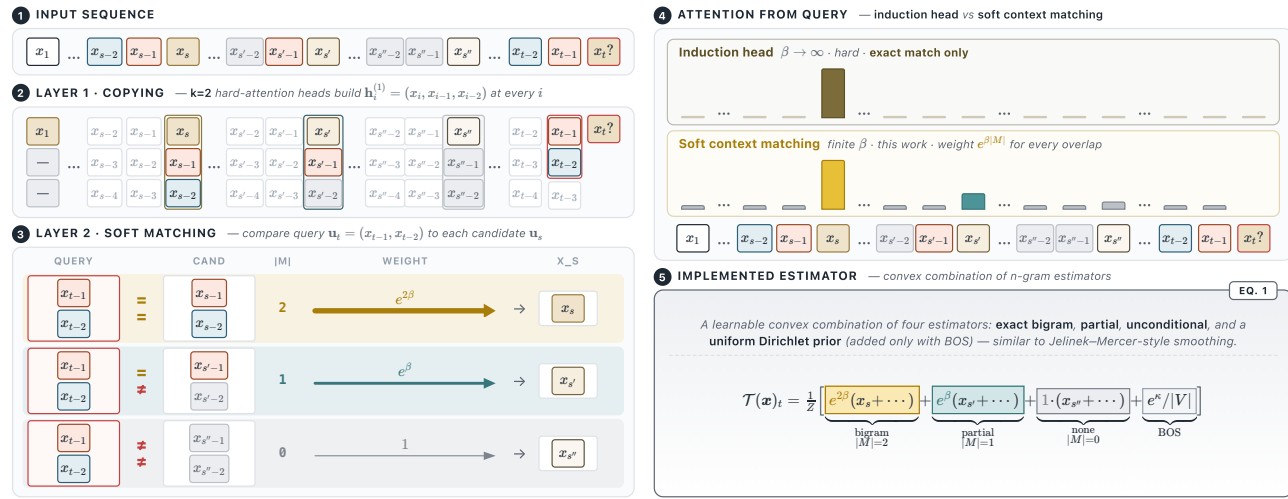

*Figure 1.* Two-layer transformer for order-$k$ Markov chains ($k=2$). **Layer 1**: Copy heads build representations $\boldsymbol{h}_i^{(1)} = (x_i, x_{i-1}, x_{i-2})$. **Layer 2**: Compares query context at $t$ with candidate contexts at positions $s, s', s''$. Vertical lines show position-wise comparison ($=$ match, $\neq$ mismatch). Attention weights scale as $e^{\beta|M|}$ (arrow thickness). Successor tokens are aggregated into a probability distribution.

where $\boldsymbol{I}_{|V|}$ is the $|V| \times |V|$ identity matrix and $\boldsymbol{S} \in \mathbb{R}^{(k+1) \times (k+1)}$ is a scaled shift matrix with entries $\S_{r,r+1} = \beta_r$ for $r \in [k]$ and zeros elsewhere. The attention matrix $\boldsymbol{W}_A^{(2)}$ has a block-shifted structure that aligns query context positions with key successor positions:

$$\boldsymbol{W}_A^{(2)} = \begin{array}{c} \\ x_i \\ x_{i-1} \\ x_{i-2} \\ \vdots \\ x_{i-k+1} \\ x_{i-k} \end{array} \begin{array}{ccccc} x_j & x_{j-1} & x_{j-2} & \cdots & x_{j-k} \\ \left(\begin{array}{ccccc} \boldsymbol{0} & \beta_1 \boldsymbol{I}_{|V|} & \boldsymbol{0} & \cdots & \boldsymbol{0} \\ \boldsymbol{0} & \boldsymbol{0} & \beta_2 \boldsymbol{I}_{|V|} & \cdots & \boldsymbol{0} \\ \boldsymbol{0} & \boldsymbol{0} & \boldsymbol{0} & \cdots & \boldsymbol{0} \\ \vdots & \ddots & \ddots & \ddots & \vdots \\ \boldsymbol{0} & \cdots & \boldsymbol{0} & \cdots & \beta_k \boldsymbol{I}_{|V|} \\ \boldsymbol{0} & \cdots & \boldsymbol{0} & \cdots & \boldsymbol{0} \end{array}\right) \end{array}$$

This structure ensures that the attention score computes $\sum_{r=1}^{k} \beta_r x_{i-r+1}^\top x_{j-r}$, comparing the query context $u_i$ with the key's predecessor context $u_{j-1}$. **Computing attention.** For a query at position $T$, the layer-2 score decomposes as

$$e_{Tj}^{(2)} = \left(\boldsymbol{h}_T^{(1)}\right)^\top \boldsymbol{W}_A^{(2)} \boldsymbol{h}_j^{(1)} + r_{T-j}^{(2)}.$$

For valid contexts $j \geq k'$, the positional term vanishes,

$$e_{Tj}^{(2)} = \sum_{r=1}^{k} \beta_r \, \mathbb{I}\{x_{T-r+1} = x_{j-r}\} = |\boldsymbol{\beta}|_{M_j^{(T)}}.$$

For the early positions $j < k'$, the RPE suppresses all scores by sending $\delta_1^{(2)} \to \infty$, except possibly the BOS position $j = 1$. When a BOS token is present:

$$e_{T1}^{(2)} = \sum_{r=1}^{k} \beta_r \, x_{T-r+1}^\top x_{\text{BOS}} + \delta_2^{(2)},$$

where we set $x_{\text{BOS}} = \frac{1}{|V|} \mathbf{1}_{|V|}$. Since every component of $x_{\text{BOS}}$ equals $1/|V|$, the content-based score is query-independent: $\sum_{r=1}^{k} \beta_r \, x_{T-r+1}^\top x_{\text{BOS}} = |\boldsymbol{\beta}|_1/|V|$. Setting

$\delta_2^{(2)} = \kappa - |\boldsymbol{\beta}|_1/|V|$ gives $e_{T1}^{(2)} = |\boldsymbol{\beta}|_1/|V| + \delta_2^{(2)} = \kappa$. Hence, for all $j \in [T]$,

$$e_{Tj}^{(2)} = \begin{cases} \kappa & \text{if } j = 1 \text{ and with BOS token}, \\ |\boldsymbol{\beta}|_{M_j^{(T)}} & \text{if } j \geq k', \\ -\infty & \text{otherwise}. \end{cases}$$

Applying softmax, we obtain

$$\mathcal{A}_{Tj}^{(2)} \propto \begin{cases} \exp(\kappa) & \text{if } j = 1 \text{ and with BOS token}, \\ \exp\left(|\boldsymbol{\beta}|_{M_j^{(T)}}\right) & \text{if } j \geq k', \\ 0 & \text{otherwise}. \end{cases}$$

Grouping terms by their match masks:

$$\sum_{j=k'}^{T} \exp\left(|\boldsymbol{\beta}|_{M_j^{(T)}}\right) = \sum_{M \subseteq [k]} N_M^{(T)} \, e^{|\boldsymbol{\beta}|_M}.$$

The normalization constant is

$$Z = \sum_{M \subseteq [k]} N_M^{(T)} \, e^{|\boldsymbol{\beta}|_M} + e^\kappa \mathbb{I}\{e_{T1}^{(2)} > 0\}.$$

**Value aggregation.** The value vectors are the layer-1 outputs $\boldsymbol{h}_j^{(1)}$. The output of layer-2 head 1 at position $T$ is: $\hat{\boldsymbol{h}}_T^{(2,1)} = \sum_{j=1}^{T} \mathcal{A}_{T,j}^{(2)} \boldsymbol{h}_j^{(1)} \in \mathbb{R}^{(k+1)|V|}$. The full layer-2 stream is $\boldsymbol{h}_T^{(2)} = \text{Concat}\left(\boldsymbol{h}_T^{(1)}, \hat{\boldsymbol{h}}_T^{(2,1)}\right) \in \mathbb{R}^{2(k+1)|V|}$. Extracting the first block of $\hat{\boldsymbol{h}}_T^{(2,1)}$, we obtain for each $m \in V$

$$\hat{\boldsymbol{h}}_T^{(2,1)}[m] = \frac{e^\kappa/|V| \, \mathbb{I}\{e_{T1}^{(2)} > 0\} + \displaystyle\sum_{M \subseteq [k]} e^{|\boldsymbol{\beta}|_M} N_M^{(T)}(m)}{e^\kappa \, \mathbb{I}\{e_{T1}^{(2)} > 0\} + \displaystyle\sum_{M \subseteq [k]} e^{|\boldsymbol{\beta}|_M} N_M^{(T)}}.$$

### 3.2.3. OUTPUT LAYER

The output layer selects block $0$ of the head output $\hat{h}_T^{(2,1)}$ from the full stream $h_T^{(2)}$ as the final prediction. We define:

$$W_O = \Big( \overbrace{0_{|V|\times(k+1)|V|}}^{\text{from } h_T^{(1)}} \Big| \overbrace{\big[\,\boxed{I_{|V|}} \quad 0_{|V|\times k|V|}\,\big]}^{\text{from } \hat{h}_T^{(2,1)}} \Big) \in \mathbb{R}^{|V|\times 2(k+1)|V|}\,,$$

Applying it to the final hidden state recovers the prediction:

$$\mathcal{T}(x) = W_O\, h_T^{(2)} = \big[\,0 \mid \boxed{I_{|V|}} \ \ 0\,\big]\, h_T^{(2)} = \big(\hat{h}_T^{(2,1)}\big)_{1:|V|}\,,$$

which completes the proof. □

## 4. Relation to Classical Estimators and Smoothing Techniques

Proposition 3.1 establishes that a disentangled two-layer transformer functions as a soft context-matching estimator, with parameters $\beta \in \mathbb{R}^k$ and $\kappa \in \mathbb{R}$: $\beta$ controls context-overlap weights, while $\kappa$ controls the BOS-induced pseudo-count contribution. In this section, we first examine $\kappa$ and how to implement add-$\alpha$-type smoothing. We then characterize the smoothing induced by finite $\beta$ by drawing parallels to classical techniques, such as Jelinek-Mercer smoothing.

### 4.1. Add-$\alpha$-Type Smoothing

To mitigate the zero-frequency problem inherent in maximum likelihood estimation over sparse data, *additive* (or add-constant) smoothing is widely employed in the literature. This technique redistributes a small portion of probability mass to unseen events, ensuring strictly positive probabilities and preventing numerical instability during log-likelihood calculations. A prominent example is Laplace smoothing, which adds $+1$ to the count of each context.

Proposition 3.1 can be instantiated to show that disentangled transformers implement such smoothing when a BOS token is present.

**Corollary 4.1** (Add-$\alpha$-Type Smoothing via BOS Token). *Set $\kappa = k\beta + \ln(\alpha|V|)$ and $\beta = \beta \mathbf{1}_k$ for $\alpha = \alpha\mathbf{1}$ with $\alpha > 0$ (symmetric prior). The transformer estimator in Equation* (8) *implements add-$\alpha$-type smoothing in the limit of $\beta \to \infty$:*

$$\lim_{\beta \to \infty} \mathcal{T}(x)(m) = \frac{N_{u_T}^{(T)}(m) + \alpha}{N_{u_T}^{(T)} + \alpha|V|}\,. \tag{10}$$

The proof is given in Appendix D. Crucially, this add-$\alpha$ smoothing is enabled by the BOS token acting as a fixed, sequence-independent sink: its contribution to the prediction is the same for every input, independent of the context.

### 4.2. Sub-$n$-gram Interpolation Smoothing

The estimator in Equation (8) admits a representation that parallels classical *interpolation smoothing* techniques (Je-

linek, 1980; Chen & Goodman, 1999). In Jelinek–Mercer (JM) smoothing, the next-token distribution is a *fixed* convex combination of the maximum-likelihood $n$-gram models of every order $0 \le i \le k$,

$$P_{\mathrm{JM}}(m \mid u_T) = \sum_{i=0}^{k} \lambda_i\, \hat{p}_i(m \mid u_T^{(i)})\,, \quad \sum_{i=0}^{k} \lambda_i = 1\,, \tag{11}$$

where $u_T^{(i)} := (x_{T-i+1}, \ldots, x_T)$ is the contiguous length-$i$ suffix of the query context $u_T$ (with $u_T^{(0)} = ()$ the empty context, giving the unigram), $\hat{p}_i(m \mid u_T^{(i)})$ is the corresponding empirical transition probability, and the mixing weights $\lambda_i$ are set globally or tuned on held-out data. The transformer implements an analogous scheme, but interpolates over *cumulative counts* rather than contiguous suffix orders.

**Definition 4.1** (Cumulative Counts). For a fixed query position $t$ and any $M \subseteq [k]$, define *cumulative* counts $K_M^{(t)}$ and *cumulative transition* counts $K_M^{(t)}(m)$:

$$K_M^{(t)} := \#\big\{s \in [k', t] : M_s^{(t)} \supseteq M\big\}\,,$$
$$K_M^{(t)}(m) := \#\big\{s \in [k', t] : M_s^{(t)} \supseteq M,\ x_s = m\big\}\,.$$

Recall that $N_M^{(t)}$ counts a position $s$ *if and only if* the indices in $M$ are the only matching positions, i.e., $M_s^{(t)} = M$. By contrast, $K_M^{(t)}$ counts the number of times the indices in $M$ are matching, without any conditions on the indices in $[k] \setminus M$. Crucially, the $K$ counts are *nested*, any position counted in $K_{M'}^{(t)}$ is also counted in $K_M^{(t)}$ for every $M \subseteq M'$, just as every $n$-gram occurrence is also an occurrence of all its shorter suffixes, whereas the exact counts $N_M^{(t)}$ *partition* the positions by their precise match pattern. The two families are related by

$$K_M^{(t)}(m) = \sum_{[k] \supseteq S \supseteq M} N_S^{(t)}(m)\,,$$

each cumulative count aggregating the exact-mask counts over all finer patterns. Using this regrouping, we rewrite our estimator in terms of the cumulative counts:

**Lemma 4.1** (Sub-$n$-gram interpolation). *The estimator in Equation* (8) *can be equivalently rewritten with parameter $\gamma = e^\beta - 1$ when $\kappa = -\infty, \beta = (\beta, \ldots, \beta)$:* [1]

$$\mathcal{T}(x)(m) = \frac{\displaystyle\sum_{M \subseteq [k]} \gamma^{|M|}\, K_M^{(T)}(m)}{\displaystyle\sum_{M \subseteq [k]} \gamma^{|M|}\, K_M^{(T)}}\,. \tag{12}$$

This leads to the following hierarchical interpretation:

---

[1]Under the convention $0^0 := 1$, Equation (12) is a polynomial identity in $\gamma$ and remains valid at $\beta = 0$ ($\gamma = 0$), where it recovers the unigram estimator $K_\emptyset^{(T)}(m)/K_\emptyset^{(T)}$.

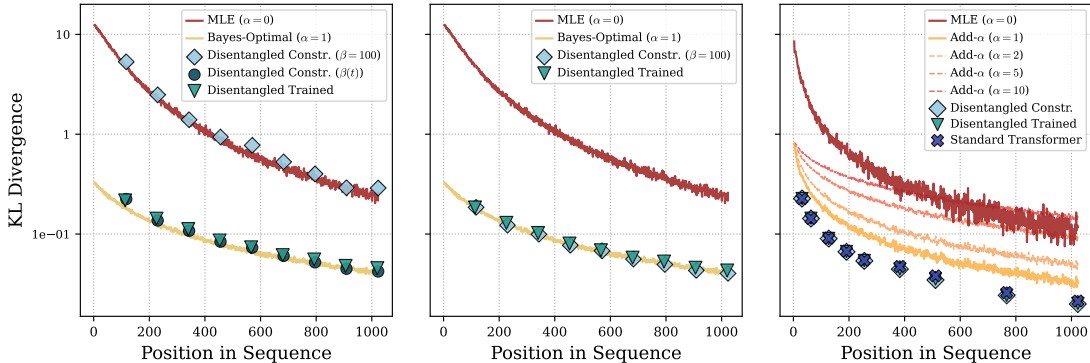

*Figure 2.* **KL divergence to the ground-truth transition distribution. Left** and **middle**: *independent* Dirichlet prior, where add-$\alpha$ smoothing is Bayes-optimal. **Left** (no BOS): with large $\beta = 100$ (blue diamonds) attention collapses to hard matching and tracks the MLE estimator (red); with adaptive $\beta(t)$ (green circles, Lemma 4.2) the transformer interpolates across context orders and approaches the Bayes-optimal estimator (yellow). **Middle** (BOS): even at large $\beta$ the BOS construction (blue) achieves near-Bayes performance via pseudo-count smoothing. In both panels, trained transformers (teal triangles) recover the smoothing behavior. **Right**: *hierarchical* Dirichlet prior ($\eta_1 = \eta_2 = 5.0$), where add-$\alpha$ smoothing is suboptimal. All models match each other and outperform add-$\alpha$ baselines.

**Corollary 4.2** (Mixtures of $n$-grams)**.** *The estimator in Equation* (8) *admits the following mixture interpretation:*

- *Chooses an order $i \in [0, k]$ with weights $\{\lambda_i^{(T)}\}$,*
- *Chooses a size-$i$ pattern $M$ ($|M| = i$) with weights $\{\lambda_M^{(T)}\}$,*
- *Predicts with the pattern model $\hat{q}_M^{(T)}(m) = \dfrac{K_M^{(T)}(m)}{K_M^{(T)}}$.*

The proofs of Lemma 4.1 and Corollary 4.2 are deferred to Appendix D. Collecting the subsets in Equation (12) by their size $|M| = i$ casts the transformer in the very same mixture form as Equation (11):

$$\mathcal{T}(\boldsymbol{x})(m) = \sum_{i=0}^{k} \lambda_i^{(T)} \, \hat{q}_i^{(T)}(m) \,,$$

$$\lambda_i^{(T)} = \frac{\gamma^i K_i^{(T)}}{\sum_{j=0}^{k} \gamma^j K_j^{(T)}} \,, \quad \hat{q}_i^{(T)}(m) = \frac{\sum_{|M|=i} K_M^{(T)}(m)}{K_i^{(T)}} \,,$$

(13)

where $K_i^{(T)} = \sum_{|M|=i} K_M^{(T)}$ and $\gamma = e^\beta - 1$ as in Lemma 4.1. These order weights $\lambda_i^{(T)}$ and order-$i$ models $\hat{q}_i^{(T)}$ are exactly the quantities of Corollary 4.2, which further splits $\hat{q}_i^{(T)} = \sum_{|M|=i} \lambda_M^{(T)} \hat{q}_M^{(T)}$ over the size-$i$ patterns. The parallel with JM is now explicit, and the two estimators differ in only two respects. *(i) Weights.* JM fixes $\lambda_i$ globally, whereas the transformer sets $\lambda_i^{(T)}$ *per sequence* from the scale factor $\gamma$ and the realized counts $K_i^{(T)}$: for $\beta > 0$, the mixture concentrates on the order-$k$ component when order-$k$ matches are present in the context, and shifts to lower orders as such matches become rare. *(ii) Per-order model.* JM's order-$i$ term is the *contiguous*-suffix MLE $\hat{p}_i(m \mid u_T^{(i)}) = K_{[i]}^{(T)}(m)/K_{[i]}^{(T)}$ with $[i] = \{1, \ldots, i\}$,

while the transformer's $\hat{q}_i^{(T)}$ averages over *all* $\binom{k}{i}$ subsets of size $i$, including non-contiguous matches. For $k = 1$ both collapse to the same two-component bigram–unigram interpolation, with specific weights in the transformer.

**Data-dependent smoothing.** For finite $\beta$, partial matches act as *structured pseudo-counts*. When the exact context $u_t$ has been rarely observed, the estimator smooths using similar contexts, where similarity is measured via an exponentiated Hamming overlap. By separating exact matches from partial matches, we obtain:

$$\mathcal{T}(\boldsymbol{x})(m) = \frac{\gamma^k K_{[k]}^{(T)}(m) + \sum\limits_{M \subsetneq [k]} \gamma^{|M|} K_M^{(T)}(m)}{\gamma^k K_{[k]}^{(T)} + \sum\limits_{M \subsetneq [k]} \gamma^{|M|} K_M^{(T)}} \,. \quad (14)$$

Since $K_{[k]}^{(T)}(m) = K_{u_t}(m)$, we can factor out $\gamma^k$ and define the *data-dependent pseudo-count*:

$$\tilde{\alpha}_m^{(T)}(\gamma) := \gamma^{-k} \sum_{M \subsetneq [k]} \gamma^{|M|} K_M^{(T)}(m). \quad (15)$$

Finally, Appendix D gives an approximate value of $\beta$ that implements add-constant smoothing.

**Lemma 4.2** ($\beta$-value for add-constant smoothing)**.** *The identity* $\mathbb{E}\left[\tilde{\alpha}_m^{(T)}(\gamma)\right] = \alpha$ *is approximately satisfied by*

$$\beta \approx \ln\left(1 + \frac{|V|}{\left(1 + \frac{\alpha |V|^{k+1}}{T-k-1}\right)^{1/k} - 1}\right).$$

**Relaxation of Katz Back-off** Back-off is itself a smoothing technique, but one built on a *precise, hard rule* rather

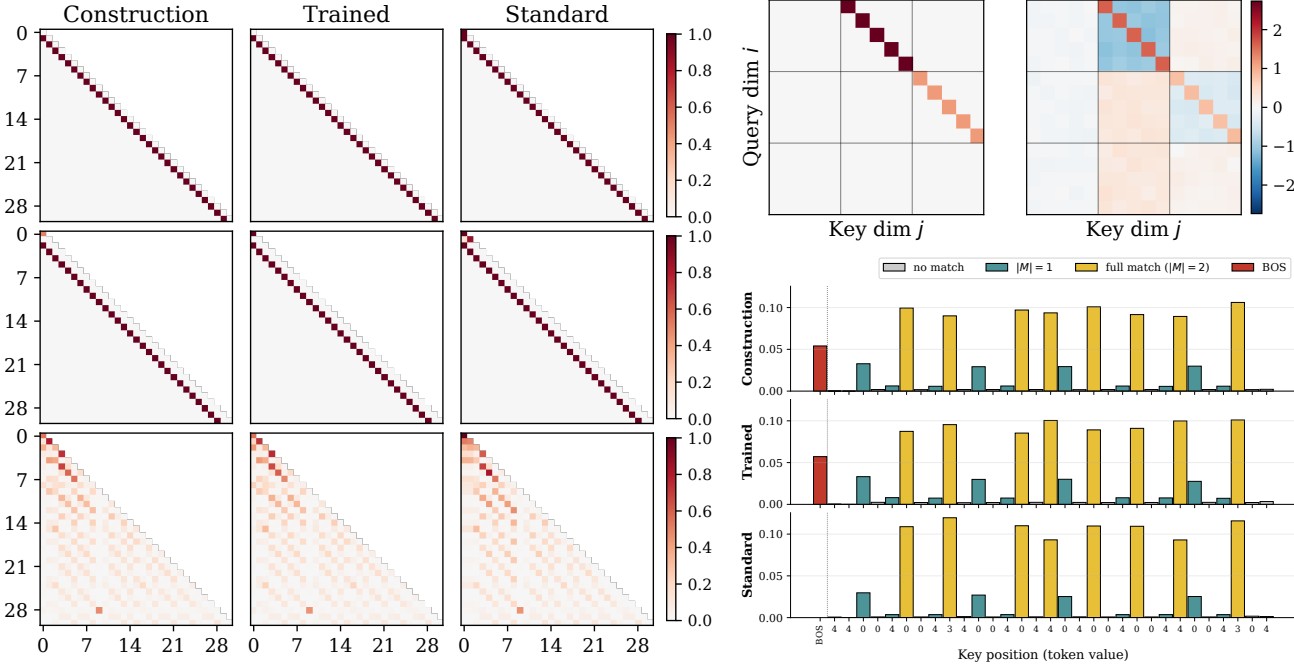

*Figure 3.* **Mechanistic evidence on the hierarchical Dirichlet task;** models: Disentangled Construction, Disentangled Trained, Standard Transformer. **Left:** attention scores; rows are Layer 1 head 1 (lag-1 copy), head 2 (lag-2 copy), and Layer 2 (matching). **Top right:** Layer-2 weight matrix $\boldsymbol{W}_A^{(2)}$ for the two disentangled models. **Bottom right:** Layer-2 attention weights at the last query position, coloured by overlap: full ($|M|=2$, yellow), partial ($|M|=1$, teal), none ($|M|=0$, grey). All three implement the same two-stage circuit: sharp lag-specific copies in Layer 1 and context matching in Layer 2 that spreads mass over full and partial matches even with BOS.

than a mixture. Like JM, Katz back-off ([Katz](), [1987]()) is built on the same contiguous-suffix models $\hat{p}_i$: it keeps the highest-order estimate whenever its count is positive and *recursively* falls back to the shorter suffix otherwise. Back-off therefore *selects* a single order per prediction, the most specific one with support, whereas Jelinek–Mercer and the transformer always *mix* all orders.

The transformer estimator Equation ([13]()) unifies the two: it is a mixture like JM, yet it performs back-off automatically. When the full context $u_T$ is unseen, $K_{[k]}^{(T)} = 0$ annihilates the top-order term and the lower orders take over, so the model "backs off" with no explicit rule. The attention-weight scale then controls the transition from Katz's hard selection rule in the $\beta \to \infty$ limit to smooth, differentiable back-off at finite $\beta$. Unlike JM and Katz back-off, it also incorporates non-contiguous matches.

## 5. Experiments

We validate our theoretical findings by comparing the KL divergence between the transformer's predictions and the ground-truth transition distribution on sequences generated from order-$k$ Markov chains, and then check that trained transformers also *implement* the predicted mechanism.

**Experimental setup.** We study order-$k$ Markov chains with $k = 2$, and vocabulary size $|V| = 5$. Transition matrices

are drawn from either an *independent* Dirichlet prior with symmetric concentration $\alpha = 1$, or a *hierarchical* Dirichlet prior with parameters $\eta_1 = \eta_2 = 5.0$. We compare three model families. **Disentangled Construction:** the disentangled two-layer transformer of Section [2]() with all weights set to the analytic values predicted by Proposition [3.1](); only the attention-weight parameter vector $\boldsymbol{\beta}$ remains trainable or is set according to Lemma [4.2](). **Disentangled Trained:** the same disentangled architecture but with all parameters optimized from initialization. **Standard Transformer:** a fully standard two-layer transformer with $d_{\text{model}} = 64$, additive residual streams, LayerNorm, an MLP block, learned relative positional biases, and learned token embeddings. All models are trained for $10^6$ iterations with Adam (lr $10^{-3}$, batch size 32) on the last-token cross-entropy loss. The BOS embedding is fixed to the neutral vector used in the construction; all other parameters remain trainable. The hierarchical task is run at different sequence lengths.

**Independent Dirichlet, no BOS.** The left panel of Figure [2]() shows the effect of $\beta$ on the estimator's behavior. At large $\beta = 100$ (blue diamonds) attention collapses to hard selection and the model implements the MLE counting estimator (red), which exhibits high KL at short sequence lengths due to sparse counts. Setting $\beta(t)$ adaptively as in Lemma [4.2]() (green circles) produces soft attention over partial context matches, the interpolation mechanism of Corollary [4.2](), and the KL approaches that of the Bayes-optimal estimator (yel-

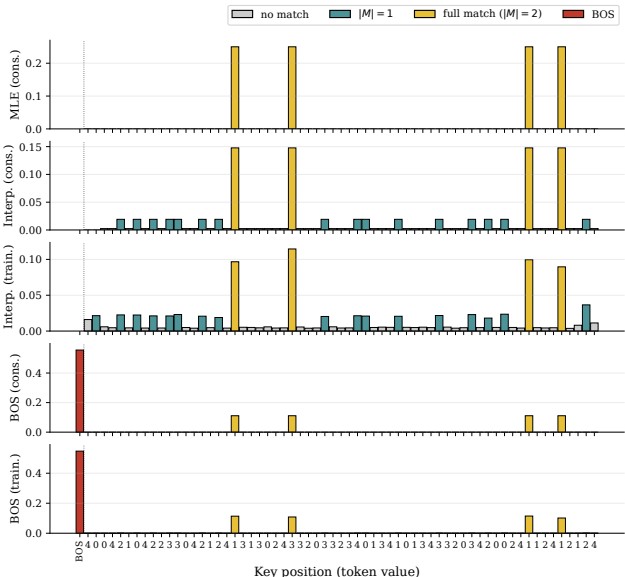

*Figure 4.* Layer-2 induction-head attention from the last query position on the independent Dirichlet task ($T=64$), for the three predicted estimators: MLE (hard $\beta$, construction); interpolation (adaptive $\beta$; construction and trained); and BOS (hard $\beta$; construction and trained). Each bar is a key position, coloured by query–key context overlap. Hard $\beta$ attends only to exact matches; finite $\beta$ spreads mass over partial matches (interpolation); BOS routes a fixed share to the BOS token. The trained transformers reproduce the construction's patterns.

low). Trained transformers (teal triangles) discover the same behavior. Weight visualizations are in App. C.1 (construction) and C.2 (trained).

**Independent Dirichlet, with BOS.** As predicted by Corollary 4.1, prepending a BOS token (middle panel of Figure 2) yields near-Bayes performance *even at large* $\beta$ (blue diamonds) regularizing the estimator without requiring soft attention. Trained BOS transformers (teal triangles) reproduce this near-Bayes behavior across all positions. Weights are in App. C.3 and C.4.

**Hierarchical Dirichlet.** The hierarchical setting is the natural testbed for our interpolation mechanism: partial matches are informative about the longer context as explained in Section 2.2. Figure 2 shows all three model families track each other closely and substantially outperform every fixed add-$\alpha$ baseline (dashed). With only two trainable scalars $\beta_1, \beta_2$, the minimal construction already matches the fully trained transformers, supporting soft context matching as the key mechanism. Additional results are in App. C.5.

**Direct mechanistic evidence.** KL divergence results show that trained transformers *behave* like our construction; here we show they also *implement* it. Figure 4 makes this concrete on the independent task: it plots the attention weights from the last query position, coloured by the query–key context overlap. The induction head realizes each of the

three predicted estimators: hard $\beta$ (MLE) attends only to exact context matches; finite $\beta$ additionally spreads mass over partial matches (interpolation); and the BOS construction attends only to the exact matches plus the BOS token (add-$\alpha$). The trained transformers reproduce the construction's attention for both the interpolation and BOS variants. The same two-stage circuit emerges on the harder hierarchical task (Figure 3): Layer 1 heads copy tokens at lag 1 and lag 2, and Layer 2 implements soft context matching. The Layer 2 weight matrix $W_A^{(2)}$ of the two disentangled models recovers the block-diagonal shift structure predicted by Proposition 3.1 (top right). Appendix E explains why label symmetry naturally favors such identity-based token comparisons. The attention weights from the last query position (bottom right), distribute mass across both full ($|M| = 2$, yellow) and partial ($|M| = 1$, teal) context matches even when the BOS is present, still implementing the interpolation predicted by Lemma 4.1, in contrast with the independent case. The standard transformer recovers the same mechanism, confirming that interpolation across partial context matches is a property of the attention rather than an artifact of the disentangled parameterization.

## 6. Conclusions

We characterize induction-head circuits as regularized estimators for in-context prediction on order-$k$ Markov chains. At finite attention-weight scale, a two-layer disentangled transformer implements a soft context-matching estimator: it aggregates successor tokens from exact and partial context matches, with weights determined by context overlap. This yields a data-dependent interpolation across context orders, analogous to Jelinek–Mercer smoothing but with weights adapted to each sequence. A complementary mechanism comes from the BOS token: by providing a sequence-independent attention target, it enables additive pseudo-counts and recovers add-$\alpha$ smoothing. Empirically, trained disentangled and standard transformers recover the predicted attention signatures. When pseudo-count smoothing is Bayes-optimal, they approach the Bayes-optimal predictor; when lower-order contexts provide structured evidence, they outperform fixed add-$\alpha$ smoothing baselines. Together, these results connect induction-head mechanisms with classical statistical smoothing, showing that transformers can *regularize* in-context estimation rather than merely count exact matches.

## Acknowledgments

This work was partially funded by an unrestricted gift from Coefficient Giving, and the grant number 212111 from the Swiss National Science Foundation. Francesco D'Angelo is supported by the Google PhD Fellowship and Oğuz Kaan Yüksel is supported by the SwissAI Fellowship.

## Impact Statement

This paper presents work whose goal is to advance the field of Machine Learning. There are many potential societal consequences of our work, none which we feel must be specifically highlighted here.

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

# A. Appendix

The appendix is organized as follows:

- In Appendix B, we discuss related work.

- In Appendix C, we present additional experiments.

- In Appendix D, we provide the proofs of Corollary 4.1, Lemmas 4.1 and 4.2 and Corollary 4.2.

- In Appendix E, we discuss label-permutation symmetry and its implications for disentangled transformers.

**Limitations.**   Our theoretical guarantees are derived for the disentangled two-layer architecture and synthetic Markov sources. The standard-transformer experiments in Figure 3 suggest that the mechanism generalizes beyond the disentangled setup, but extending the analysis to deeper stacks, large-scale models, and natural text is left for future work.

# B. Related Work

**Mechanisms for in-context learning.**   The empirical discovery of in-context learning in large language models (Brown et al., 2020) led to several complementary explanations of how transformers adapt from prompts. One line of work studies ICL through the lens of algorithm learning: transformers trained on families of regression problems can learn procedures resembling gradient descent, least squares, or higher-order optimization methods (Garg et al., 2022; Akyürek et al., 2022; Von Oswald et al., 2023a;b; Zhang et al., 2024a; Ahn et al., 2023; Fu et al., 2024), while transformers trained on discrete-token mixture-of-transition tasks can implement mirror descent to infer latent mixture weights (D'Angelo & Flammarion, 2026). Other work emphasizes statistical or Bayesian structure, viewing ICL as implicit inference over latent concepts or task parameters (Xie et al., 2022; Zhang et al., 2024b). The emergence and form of these algorithms can depend on the diversity of pretraining tasks, and transformers can also select among candidate algorithms or task families from the prompt (Raventos et al., 2023; Bai et al., 2023; Yadlowsky et al., 2023). These perspectives are complementary to ours. Rather than studying regression or generic latent-task inference, we isolate next-token prediction in Markovian sequences and ask which finite-sample estimator is implemented by the attention circuit.

**Induction heads and circuit formation.**   Mechanistic interpretability has shown that transformers can contain recognizable computational circuits (Elhage et al., 2021), with induction heads serving as a central example of a circuit for copying from repeated contexts (Olsson et al., 2022). Subsequent work has investigated how these circuits form during training, how simpler subcircuits interact before induction behavior appears, and how data properties such as burstiness, imbalance, and repetition influence their emergence (Chan et al., 2022; Singh et al., 2024; Zucchet et al., 2026). Automated circuit-discovery tools and interpretable-by-design models provide related routes for making such mechanisms explicit (Conmy et al., 2023; Friedman et al., 2023). Our analysis contributes to this line by assigning a statistical role to the induction-head computation: soft context matching does not merely copy from repeated contexts, but implements a smoothed estimator over partial matches.

**Markov chains, $n$-grams, and transformer ICL.**   Sequential probabilistic models provide a controlled setting for understanding next-token prediction. Recent theory gives sample-complexity bounds for next-token prediction on Markovian data (Yüksel et al., 2025; Yüksel & Flammarion, 2025b;a), while mechanistic studies show that transformers trained on bigrams or Markov chains develop induction-like mechanisms for estimating transition probabilities in context (Bietti et al., 2023; Edelman et al., 2024; Nichani et al., 2024). This picture has been extended to higher-order Markov chains (Chen et al., 2024; Rajaraman et al., 2024) and causal-structure selection (D'Angelo et al., 2025). For continuous autoregressive sequences, trained transformers can first infer a linear transition map in context and then apply it for prediction, with one-layer linear models implementing a gradient-descent step in structured settings (Sander et al., 2024); for noisy linear dynamical systems, an optimal single linear-attention construction similarly corresponds to one gradient-descent step on a window-size-one autoregression objective, with larger windows connected empirically to generalized preconditioned conjugate gradient methods (Vladarean et al., 2026). Other related extensions include loss-landscape analyses (Makkuva et al., 2024) and near-stationary $n$-gram solutions (Varre et al., 2025). Our work differs in emphasis: instead of treating the learned predictor as hard transition-count estimation, we characterize the soft context-matching estimator induced by attention and show that it interpolates among exact and partial context matches.

**Transformers as sequential models.** A broader theoretical literature studies the representational power and limitations of transformers on sequence tasks. Transformers are universal approximators of sequence-to-sequence functions under suitable assumptions (Yun et al., 2020) and can be computationally powerful models of sequence processing (Pérez et al., 2021; Sanford et al., 2024). For language-model-like distributions, sparse-attention transformers can represent $n$-gram models exactly (Svete & Cotterell, 2024), though other sequential families such as hidden Markov models can expose limitations relative to recurrent architectures (Hu et al., 2024). Related work also finds interpretable belief-state structure inside transformers trained on hidden-state inference problems (Shai et al., 2024). For linear attention, asymptotic analyses provide exact characterizations of ICL in high-dimensional limits (Lu et al., 2025). At a more phenomenological level, $n$-gram statistics can approximate some transformer predictions, but this does not by itself explain how the model selects the relevant rule from context (Nguyen, 2024). These results motivate studying not only what sequential distributions transformers can represent, but also which estimators their attention mechanisms favor in finite-context regimes.

**Classical smoothing and our position.** Classical $n$-gram language models confronted the same sparsity problem that appears in finite-context ICL: high-order contexts are informative when observed often, but unreliable when their counts are small. Smoothing methods such as interpolation, backoff, and hierarchical Dirichlet models address this bias-variance tradeoff by borrowing strength from lower-order or prior distributions (Jelinek, 1980; Katz, 1987; Chen & Goodman, 1999; MacKay & Peto, 1995). Recent unbounded $n$-gram models show that count-based methods remain relevant even at modern data scales (Liu et al., 2024). Our contribution is to connect these classical estimators to transformer circuits: soft context matching implements interpolation through attention weights, while the BOS token supplies additive pseudo-counts.

## C. Additional Experiments

In this appendix, we provide detailed visualizations of the attention patterns and internal representations for both the BOS (Beginning-Of-Sequence) and no-BOS constructions described in the main text, as well as for trained transformer models. These visualizations serve two purposes: (i) comparing the learned weights and attention patterns with our theoretical constructions, and (ii) providing a clear visualization of the interpolation mechanism via the induction head plots, where the contrast between soft (partial context matching) and hard ($n$-gram matching) mechanisms is clearly visible. Unless otherwise stated, all visualizations in this section use a single sequence of length $T = 64$ drawn from the independent Dirichlet prior ($\alpha = 1$); trained models use the converged seed.

### C.1. No-BOS Construction Visualizations

We visualize the no-BOS construction in two $\beta$ regimes that share the *same* weights but differ in Layer 2: (i) a *fixed* large $\beta$, which collapses Layer 2 to hard $n$-gram matching, and (ii) the *adaptive* schedule $\beta(t)$ of Lemma 4.2, which softens Layer 2 into interpolation across context orders. The $R_A$ vectors and the Layer 2 weight matrix $W_A^{(2)}$ are identical across regimes (up to the overall $\beta$ scale of $W_A^{(2)}$), so we repeat them for clarity; the regimes differ only in the Layer 2 post-softmax attention weights and the induction-head bar plot. Throughout this appendix we show Layer 1 attention *after* softmax and Layer 2 attention *before* softmax (scores); the sole exception is this construction comparison, where we show the Layer 2 *post-softmax weights*, since that is precisely where the two regimes differ (the pre-softmax scores are $\beta$-invariant up to scale).

FIXED LARGE $\beta$: HARD $n$-GRAM MATCHING

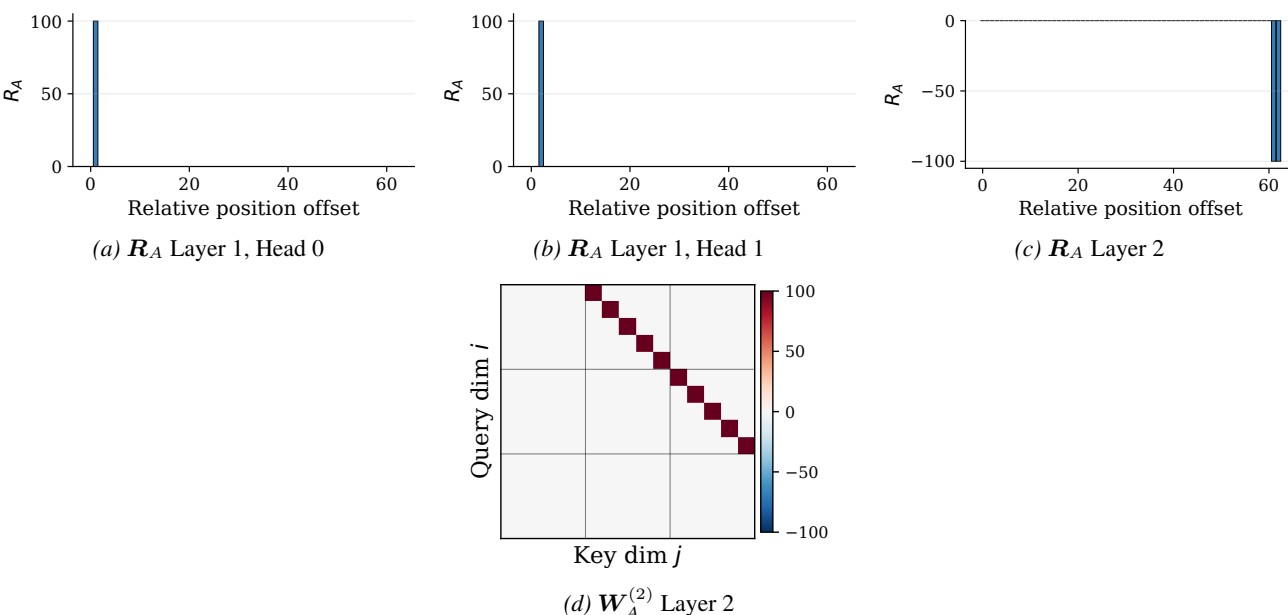

*(a)* $\boldsymbol{R}_A$ Layer 1, Head 0   *(b)* $\boldsymbol{R}_A$ Layer 1, Head 1   *(c)* $\boldsymbol{R}_A$ Layer 2

*(d)* $\boldsymbol{W}_A^{(2)}$ Layer 2

*Figure 5.* No-BOS construction: relative positional encodings $\boldsymbol{R}_A$ (Layer 1 heads copy at lag 1 and lag 2) and the Layer 2 weight matrix $\boldsymbol{W}_A^{(2)}$ with its block-diagonal shift structure. These weights are shared with the adaptive regime below.

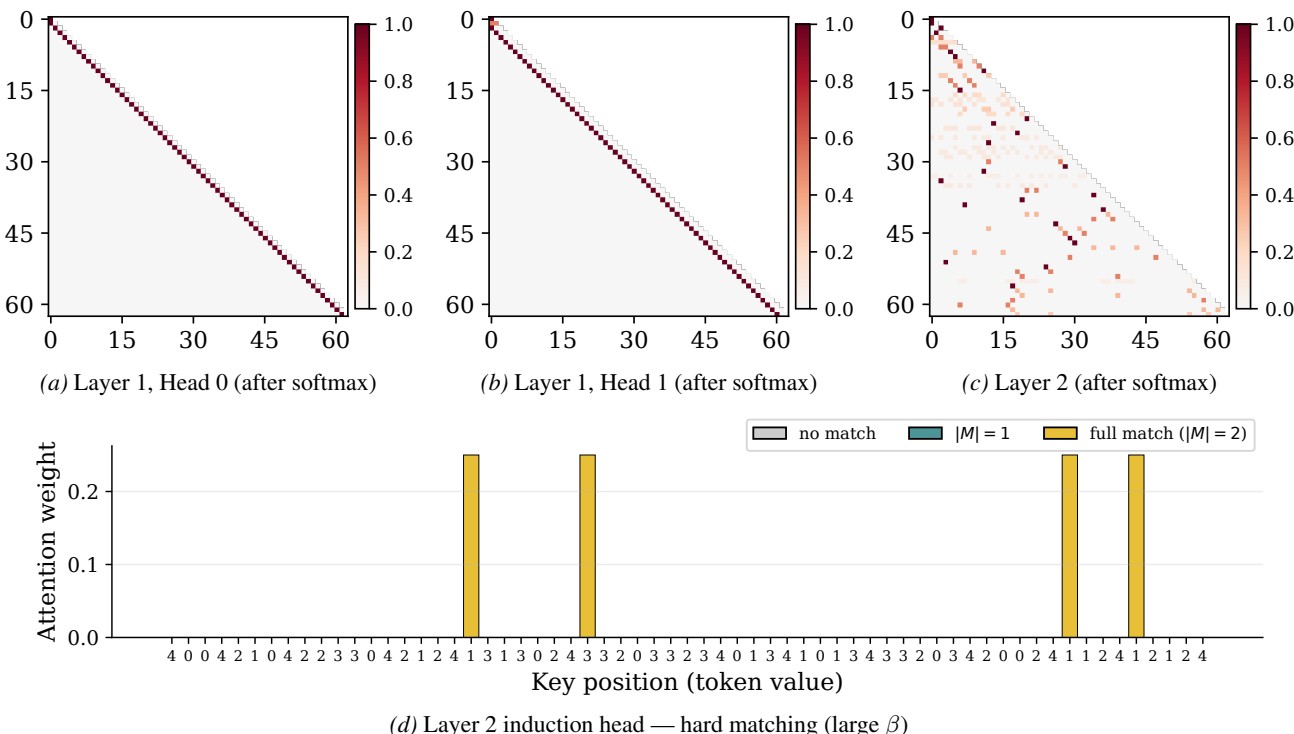

*(a)* Layer 1, Head 0 (after softmax)   *(b)* Layer 1, Head 1 (after softmax)   *(c)* Layer 2 (after softmax)

*(d)* Layer 2 induction head — hard matching (large $\beta$)

*Figure 6.* No-BOS construction, **fixed large** $\beta$: Layer 1 post-softmax copy attention and the Layer 2 post-softmax attention weights, which concentrate on exact $n$-gram matches. The induction-head bar (bottom) shows attention from the last query position, coloured by degree of context match.

ADAPTIVE $\beta(t)$: INTERPOLATION SMOOTHING

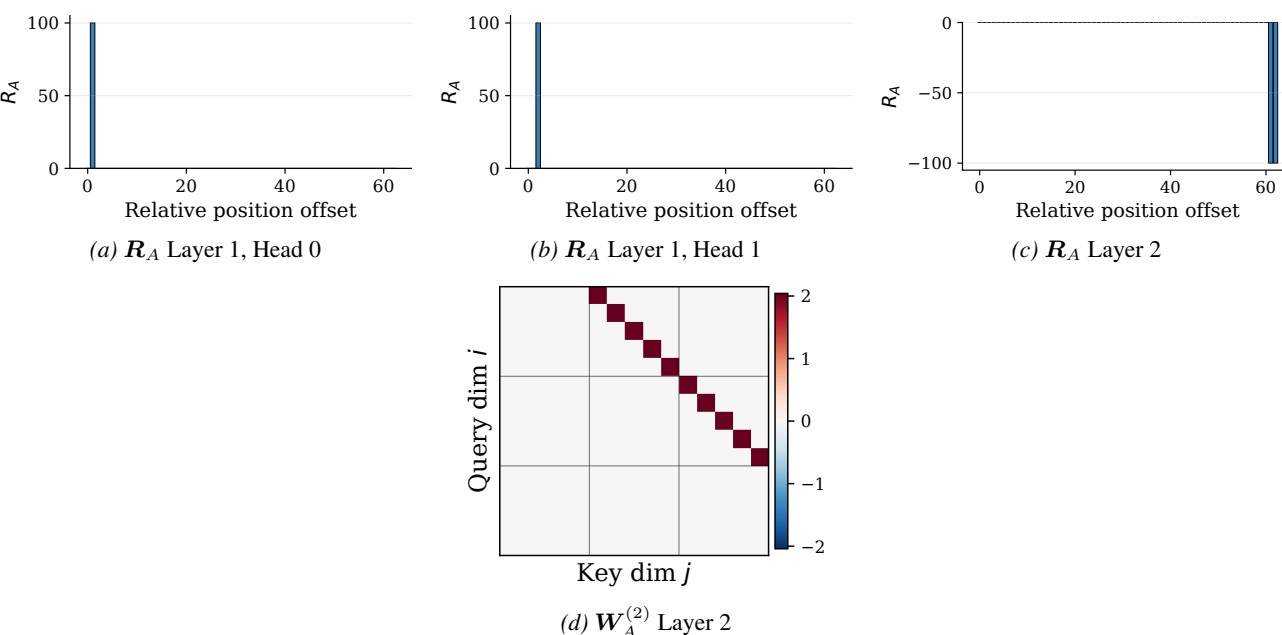

*Figure 7.* No-BOS construction: the same $\boldsymbol{R}_A$ and $\boldsymbol{W}_A^{(2)}$ as the fixed regime above (repeated for reference; $\boldsymbol{W}_A^{(2)}$ differs only by the overall $\beta$ scale).

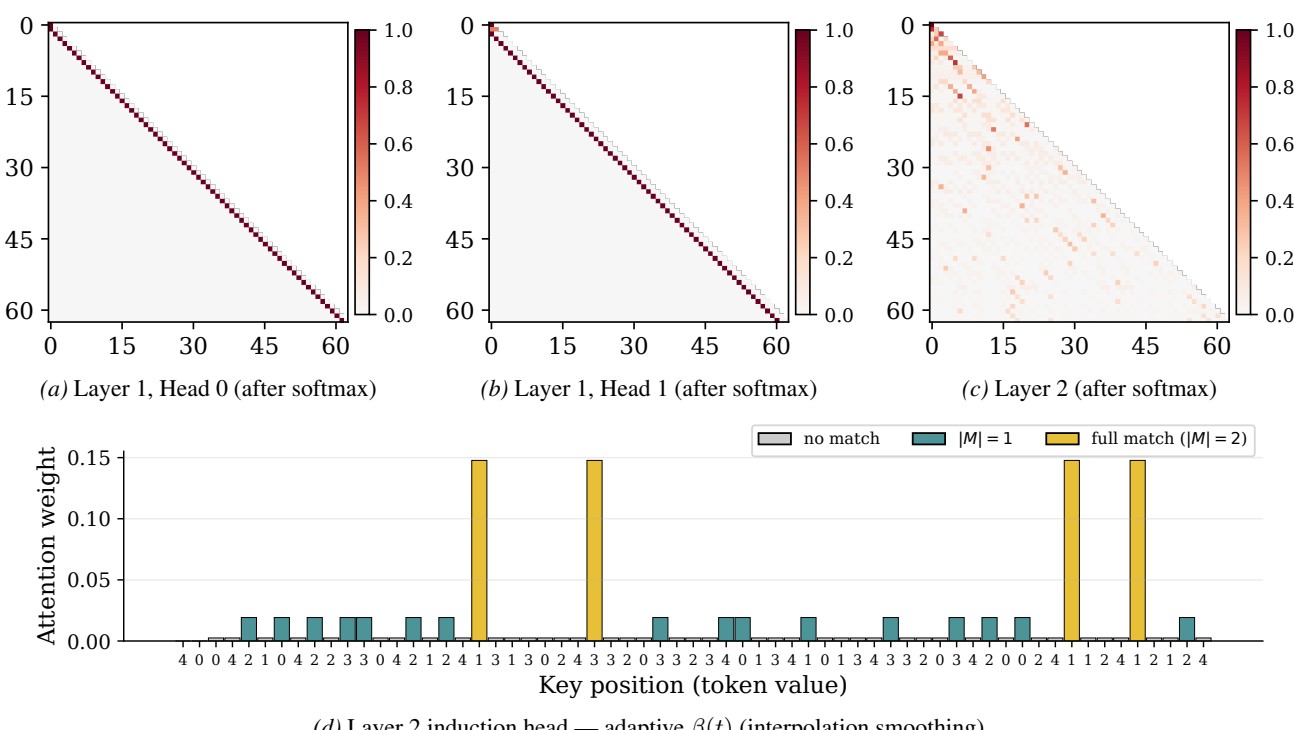

*Figure 8.* No-BOS construction, **adaptive** $\beta(t)$: identical Layer 1 copy attention, but the Layer 2 post-softmax attention weights now spread over partial matches (the interpolation of Lemma 4.1), and the induction bar distributes mass across full and partial matches.

## C.2. No-BOS Trained Model Visualizations

We visualize the learned weights and attention patterns from a transformer trained without the BOS token (Layer 1 initialized to hard copy heads but trainable, Layer 2 fully unconstrained). The similarity between the learned weights and the construction (Section C.1) validates the theoretical analysis.

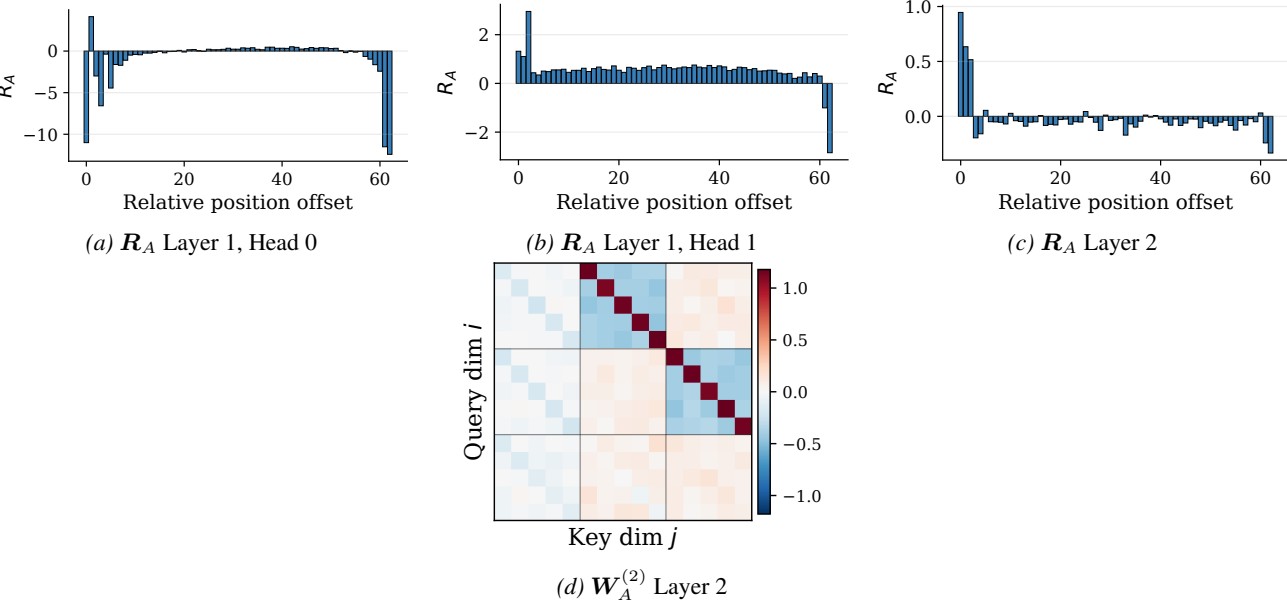

*Figure 9.* No-BOS trained model: learned relative positional encodings $\boldsymbol{R}_A$ and Layer 2 weight matrix $\boldsymbol{W}_A^{(2)}$. The trained model recovers the lag-specific copy structure in Layer 1 and the block-diagonal shift structure in $\boldsymbol{W}_A^{(2)}$.

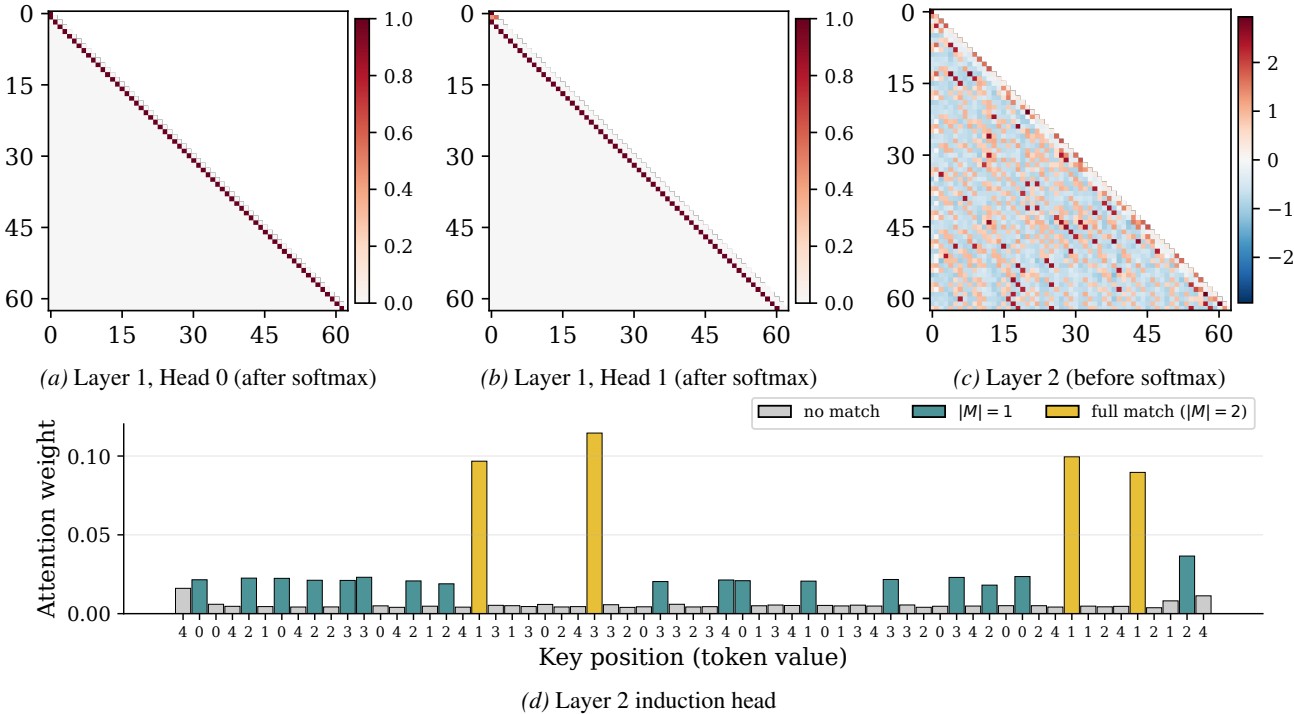

*Figure 10.* No-BOS trained model: Layer 1 post-softmax copy attention, Layer 2 pre-softmax scores, and the Layer 2 induction-head attention. The trained model implements soft context matching, distributing attention over both full and partial matches.

## C.3. BOS Construction Visualizations

We visualize the weights and attention patterns from our BOS construction. The BOS token provides a dedicated attention target that contributes pseudo-counts, enabling add-$\alpha$-style smoothing even when context matching is sharp.

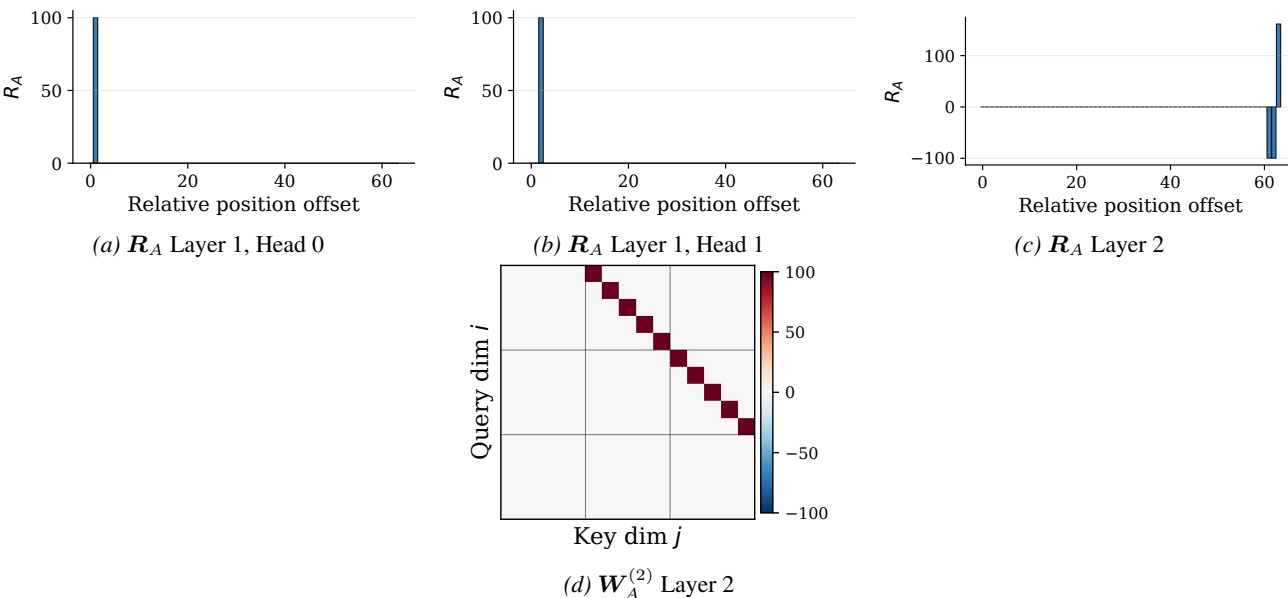

*(a) $\boldsymbol{R}_A$ Layer 1, Head 0*  *(b) $\boldsymbol{R}_A$ Layer 1, Head 1*  *(c) $\boldsymbol{R}_A$ Layer 2*

*(d) $\boldsymbol{W}_A^{(2)}$ Layer 2*

*Figure 11.* BOS construction: relative positional encodings $\boldsymbol{R}_A$ (Layer 2 $\boldsymbol{R}_A$ carries the BOS pseudo-count boost at the top offset and the first-$k$ mask) and the Layer 2 weight matrix $\boldsymbol{W}_A^{(2)}$.

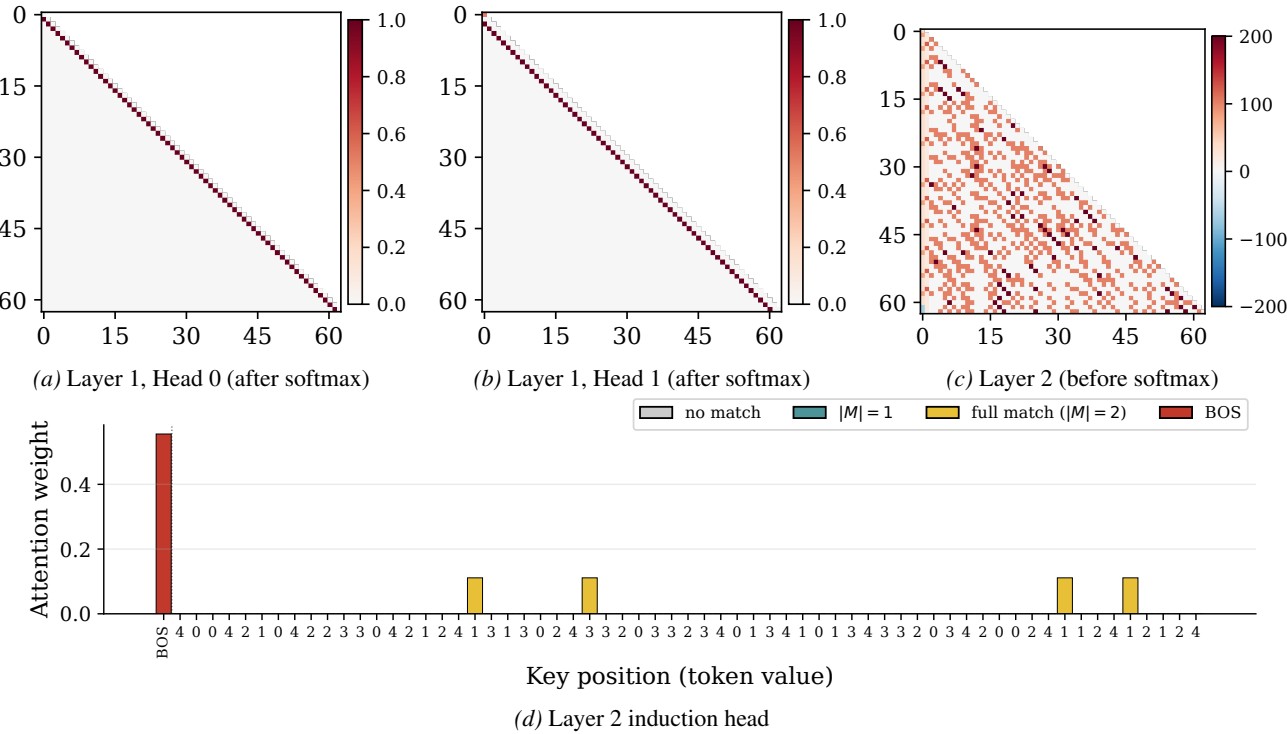

*(a) Layer 1, Head 0 (after softmax)*  *(b) Layer 1, Head 1 (after softmax)*  *(c) Layer 2 (before softmax)*

*(d) Layer 2 induction head*

*Figure 12.* BOS construction: Layer 1 post-softmax copy attention, Layer 2 pre-softmax scores, and the Layer 2 induction-head attention. The BOS pseudo-count lets the model attend sharply to exact matches while remaining well-regularized.

## C.4. BOS-Trained Model Visualizations

We visualize the learned weights and attention patterns from a transformer trained with the BOS token. Comparing to the BOS construction (Section C.3) shows how closely the trained model recovers the theoretical construction.

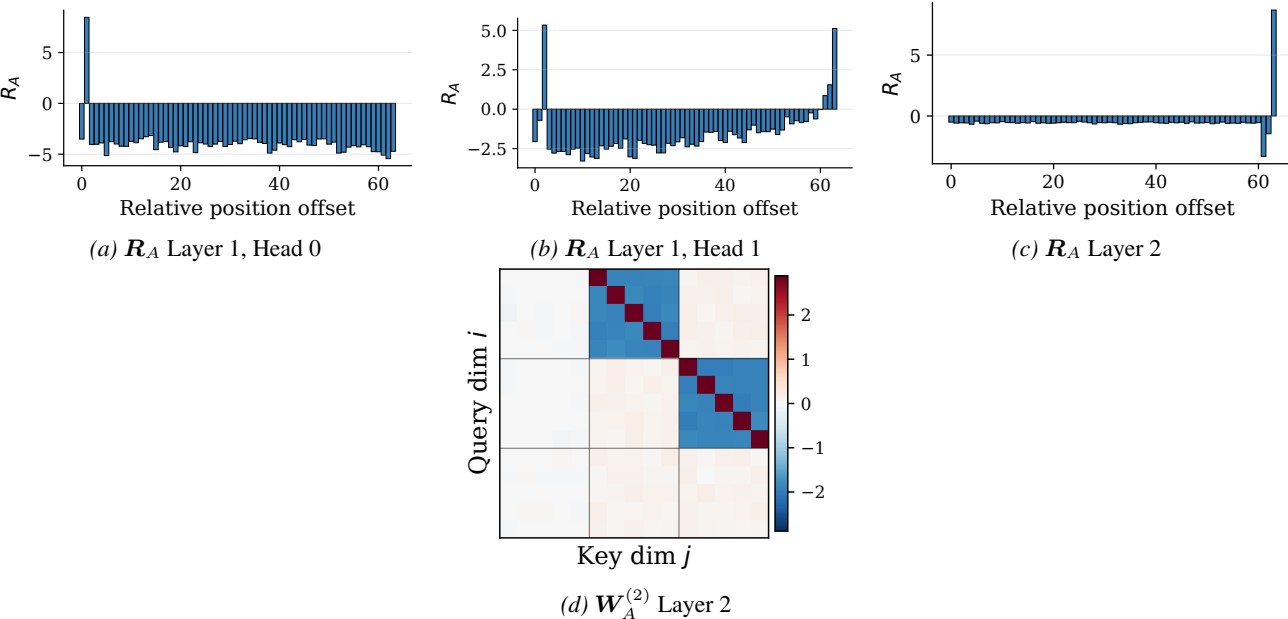

*(a) $\boldsymbol{R}_A$ Layer 1, Head 0*  *(b) $\boldsymbol{R}_A$ Layer 1, Head 1*  *(c) $\boldsymbol{R}_A$ Layer 2*

*(d) $\boldsymbol{W}_A^{(2)}$ Layer 2*

*Figure 13.* BOS trained model: learned relative positional encodings $\boldsymbol{R}_A$ and Layer 2 weight matrix $\boldsymbol{W}_A^{(2)}$, recovering the construction's structure.

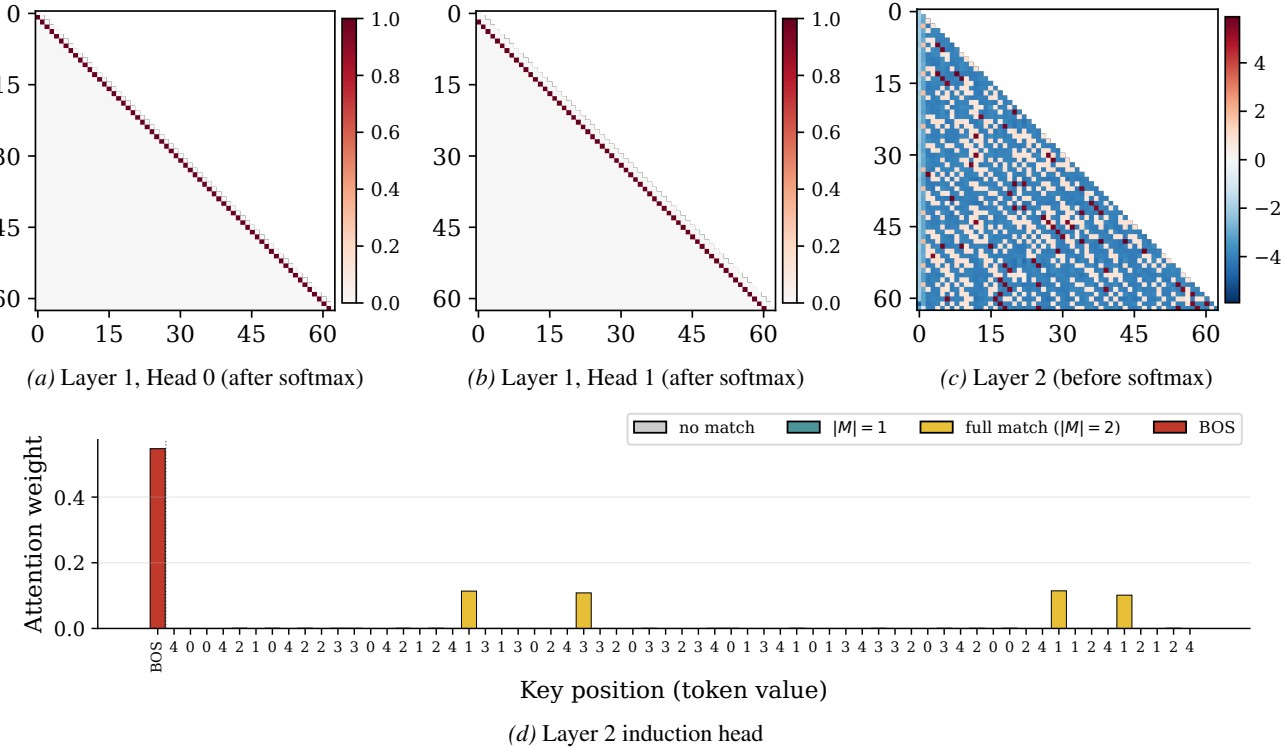

*(a) Layer 1, Head 0 (after softmax)*  *(b) Layer 1, Head 1 (after softmax)*  *(c) Layer 2 (before softmax)*

*(d) Layer 2 induction head*

*Figure 14.* BOS trained model: Layer 1 post-softmax copy attention, Layer 2 pre-softmax scores, and the Layer 2 induction-head attention, demonstrating the learned soft context-matching mechanism.

## C.5. Hierarchical Dirichlet Prior: Extended Mechanistic Visualizations

The main text establishes the hierarchical-prior results at $T{=}32$: Figure 3 shows that the BOS construction, the trained disentangled transformer, and the standard transformer all implement the same two-stage interpolation circuit, and the right panel of Figure 2 shows that all three outperform every add-$\alpha$ baseline. Here we provide the same mechanistic evidence at a *longer* sequence length, $T{=}128$, with each component shown separately—Layer 2 weight matrix, Layer 2 attention, Layer 1 copy heads, and the induction-head decomposition—rather than compressed into a single panel. The setup matches Section 5 (order-2 chains, $|V|{=}5$, $\eta_1{=}\eta_2{=}5.0$): models are trained across nine sequence lengths $T \in \{32, 64, 128, 192, 256, 384, 512, 768, 1024\}$ and 3 seeds, and we visualize the converged $T{=}128$ models. Recall that for the construction Layer 1 uses frozen hard copy heads and Layer 2 keeps only the two per-lag scalars $\beta_1, \beta_2$ trainable (no closed form is available under this prior), whereas the trained disentangled and standard models optimize all parameters.

**Layer 2 weight matrix.** Figure 15 shows $\boldsymbol{W}_A^{(2)}$ for the two disentangled models. The construction (left) has the block-diagonal structure predicted by Prop. 3.1—two shift blocks scaled by the learned $\beta_1, \beta_2$—and the trained transformer (right) recovers the same structure, confirming that gradient descent finds the predicted parameterization at $T{=}128$ just as at $T{=}32$.

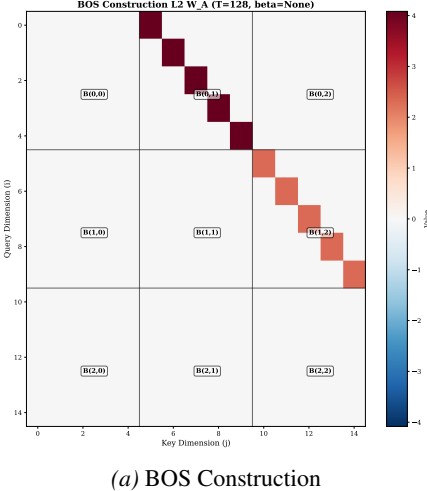

*(a)* BOS Construction

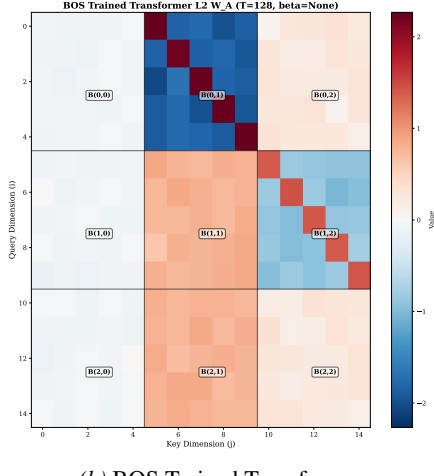

*(b)* BOS Trained Transformer

*Figure 15.* Layer 2 weight matrix $\boldsymbol{W}_A^{(2)}$ at $T = 128$ for the hierarchical Dirichlet task. Both models exhibit the block-diagonal shift structure predicted by the theory.

**Attention patterns.** Figure 16 (Layer 2) and Figure 17 (Layer 1 copy heads) show the attention for all three model families. Layer 1 head 0 copies at lag 1 and head 1 at lag 2—hard by design in the construction and learned nearly identically by the trained disentangled and standard models—while Layer 2 realizes the induction-head pattern in every case, each query attending to keys with matching preceding context.

**Induction-head decomposition.** Figure 18 plots, for the last query position, the attention weight on each key position colored by degree of context match. All three models place the most mass on full matches yet allocate non-negligible mass to partial matches—the interpolation across context orders predicted by Lemma 4.1, and the reason they improve over fixed add-$\alpha$ smoothing under the hierarchical prior.

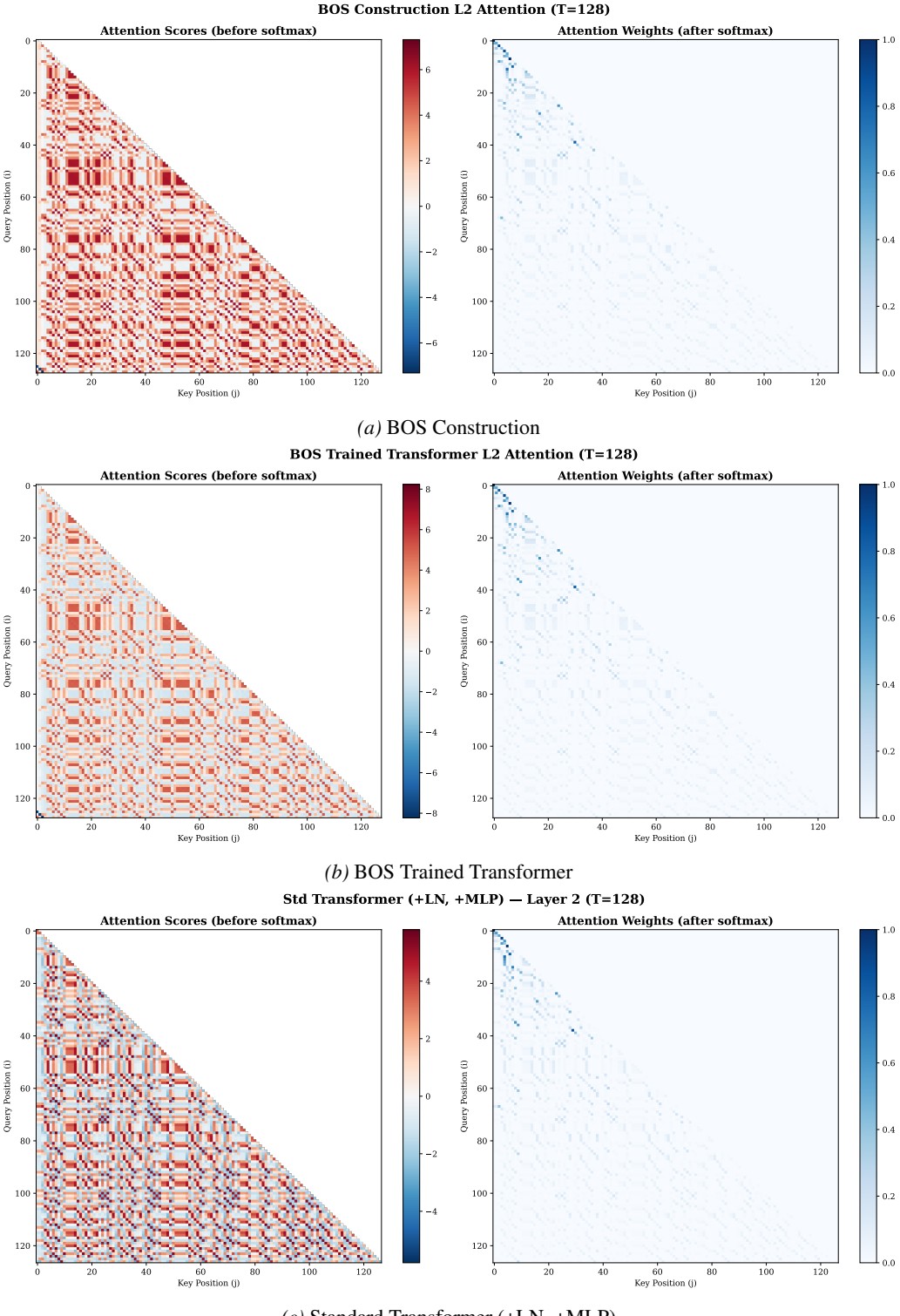

*(a)* BOS Construction

*(b)* BOS Trained Transformer

*(c)* Standard Transformer (+LN, +MLP)

*Figure 16.* Layer 2 attention at $T = 128$ for the hierarchical Dirichlet task (left: scores before softmax; right: weights after softmax). All models exhibit the induction head pattern, attending to positions with matching context.

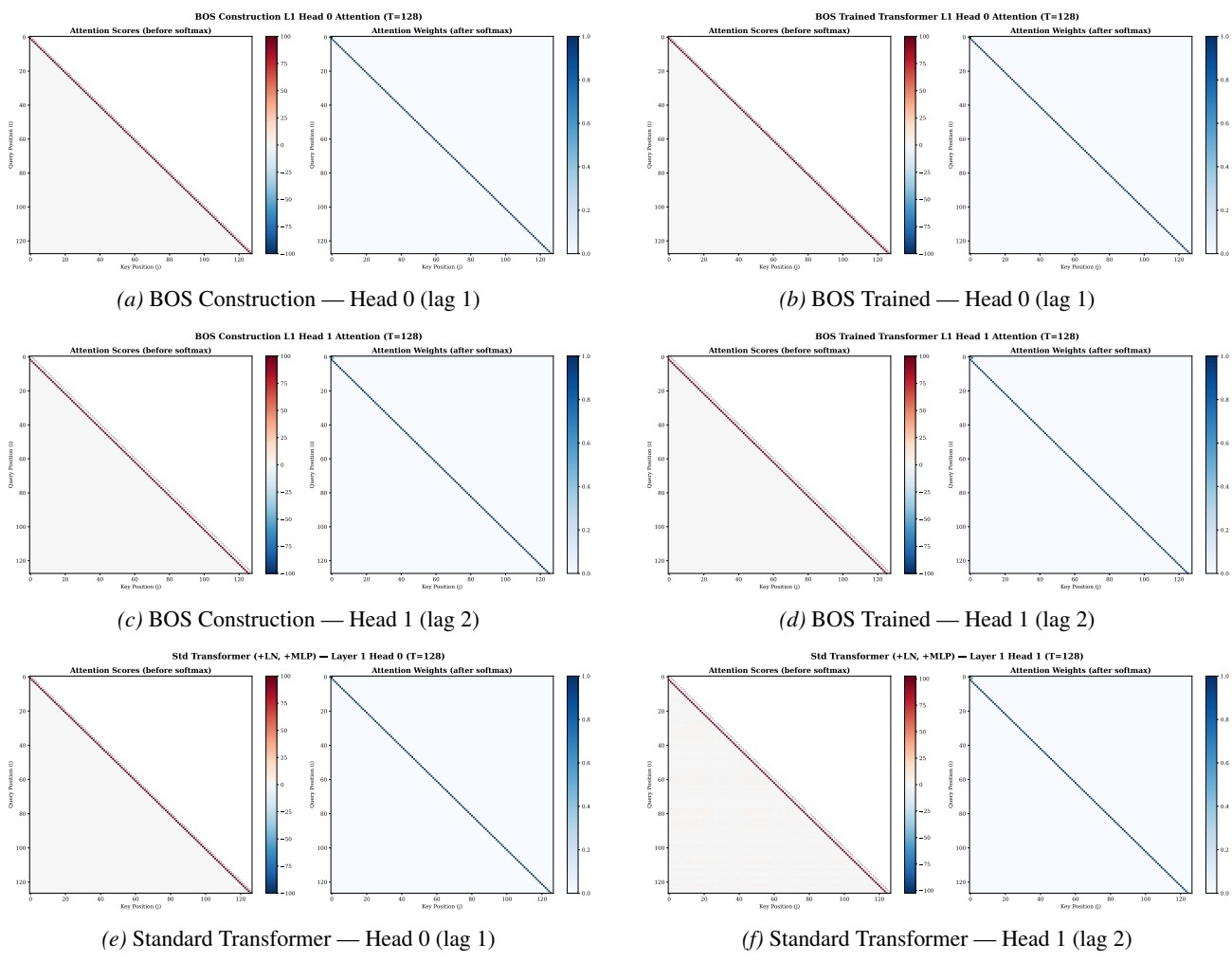

*(a)* BOS Construction — Head 0 (lag 1)

*(b)* BOS Trained — Head 0 (lag 1)

*(c)* BOS Construction — Head 1 (lag 2)

*(d)* BOS Trained — Head 1 (lag 2)

*(e)* Standard Transformer — Head 0 (lag 1)

*(f)* Standard Transformer — Head 1 (lag 2)

*Figure 17.* Layer 1 copy head attention at $T = 128$ for the hierarchical Dirichlet task. Head 0 copies the token at lag 1 and Head 1 at lag 2. All three models recover the same sharp positional attention.

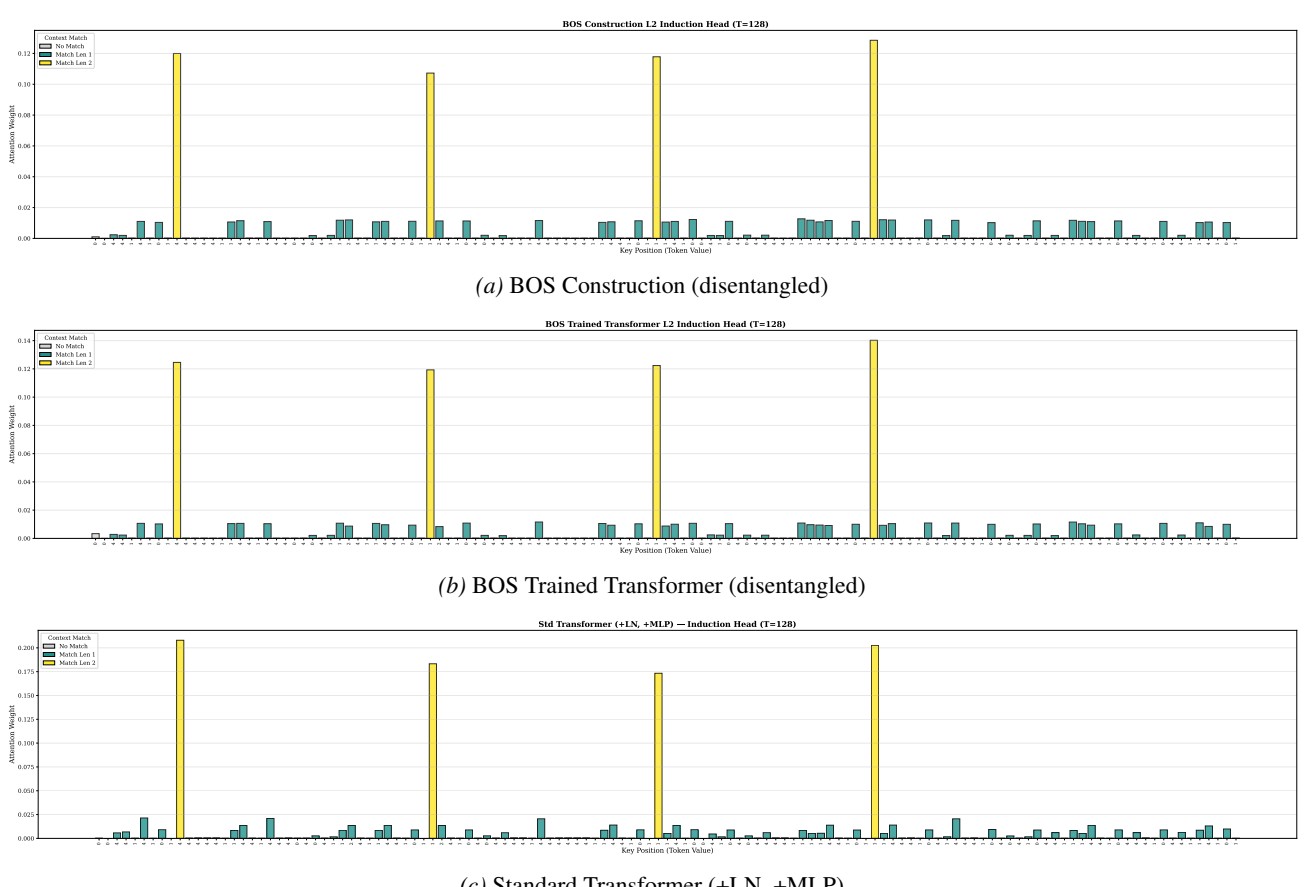

*(a)* BOS Construction (disentangled)

*(b)* BOS Trained Transformer (disentangled)

*(c)* Standard Transformer (+LN, +MLP)

*Figure 18.* Induction head analysis at $T = 128$ for the hierarchical Dirichlet task: attention weights from the last query position, colored by degree of context match. All three models attend to full matches as well as partial matches, consistent with the interpolation across context orders predicted by the theory.

**Disentangled vs. standard architecture.** The standard-transformer panels above (Figure 16(c), Figure 17(e,f), and Figure 18(c)) reproduce the same two-stage mechanism, so it is not an artifact of the disentangled parameterization. The disentangled model merely exposes the circuit directly through one-hot embeddings and concatenated residual streams; a standard transformer (learned embeddings, $d_{\text{model}}{=}64$, LayerNorm, MLP) realizes the same computation through its value matrix for copying and its query–key dot product for context matching.

# D. Proofs for Smoothing and Interpolation

This appendix contains the proofs of the results in Section 4: the add-$\alpha$ smoothing corollary (Corollary 4.1), the sub-$n$-gram interpolation lemma (Lemma 4.1), the mixture corollary (Corollary 4.2), and the optimal $\beta$ lemma (Lemma 4.2).

## D.1. Proof of Corollary 4.1

**Corollary 4.1** (Add-$\alpha$-Type Smoothing via BOS Token). *Set* $\kappa = k\beta + \ln(\alpha|V|)$ *and* $\boldsymbol{\beta} = \beta\mathbf{1}_k$ *for* $\boldsymbol{\alpha} = \alpha\mathbf{1}$ *with* $\alpha > 0$ *(symmetric prior). The transformer estimator in Equation* (8) *implements add-$\alpha$-type smoothing in the limit of* $\beta \to \infty$:

$$\lim_{\beta \to \infty} \mathcal{T}(\boldsymbol{x})(m) = \frac{N_{u_T}^{(T)}(m) + \alpha}{N_{u_T}^{(T)} + \alpha|V|}. \tag{10}$$

*Proof.* Starting from the estimator in Equation (8) with $\boldsymbol{\beta} = \beta\mathbf{1}_k$ and $\kappa = k\beta + \ln|\boldsymbol{\alpha}|_1 = k\beta + \ln(\alpha|V|)$:

$$\mathcal{T}(\boldsymbol{x})(m) = \frac{e^\kappa/|V| + \displaystyle\sum_{M \subseteq [k]} e^{\beta|M|} N_M^{(T)}(m)}{e^\kappa + \displaystyle\sum_{M \subseteq [k]} e^{\beta|M|} N_M^{(T)}}.$$

Since $e^\kappa = e^{k\beta} \cdot \alpha|V|$, the BOS numerator term is $e^{k\beta} \cdot \alpha|V|/|V| = e^{k\beta}\alpha$.

In the sum over $M \subseteq [k]$, each term carries weight $e^{\beta|M|}$. As $\beta \to \infty$, the dominant contribution comes from $M = [k]$ (exact matches), which has weight $e^{k\beta}$. All terms with $|M| < k$ are $o(e^{k\beta})$:

$$\sum_{M \subseteq [k]} e^{\beta|M|} N_M^{(T)}(m) = e^{k\beta} N_{[k]}^{(T)}(m) + o(e^{k\beta}).$$

Since $N_{[k]}^{(T)}(m) = N_{u_T}^{(T)}(m)$ (exact context match counts), we obtain:

$$\text{numerator} = e^{k\beta}\alpha + e^{k\beta} N_{u_T}^{(T)}(m) + o(e^{k\beta}),$$
$$\text{denominator} = e^{k\beta}\alpha|V| + e^{k\beta} N_{u_T}^{(T)} + o(e^{k\beta}).$$

Dividing both by $e^{k\beta}$ and taking $\beta \to \infty$:

$$\lim_{\beta \to \infty} \mathcal{T}(\boldsymbol{x})(m) = \frac{\alpha + N_{u_T}^{(T)}(m)}{\alpha|V| + N_{u_T}^{(T)}}.$$

$\square$

## D.2. Proofs of Lemmas 4.1 and 4.2 and Corollary 4.2

Recall that $K_M^{(t)}$ counts the number of times the indices in $M$ are matching, whereas, $N_M^{(t)}$ counts if *only* the indices in $M$ are matching. These two quantities are related by the following identity:

$$K_M^{(t)}(m) = \sum_{[k] \supseteq S \supseteq M} N_S^{(t)}(m).$$

Based on the Principle of Inclusion-Exclusion, we can revert the identity as follows:

$$N_M^{(t)}(m) = \sum_{[k] \supseteq S \supseteq M} (-1)^{|S|-|M|} K_S^{(t)}(m).$$

Using these relationships, we prove Lemma 4.1.

**Lemma 4.1** (Sub-$n$-gram interpolation)**.** *The estimator in Equation* (8) *can be equivalently rewritten with parameter* $\gamma = e^\beta - 1$ *when* $\kappa = -\infty, \boldsymbol{\beta} = (\beta, \ldots, \beta)$: [2]

$$\mathcal{T}(\boldsymbol{x})(m) = \frac{\displaystyle\sum_{M \subseteq [k]} \gamma^{|M|} K_M^{(T)}(m)}{\displaystyle\sum_{M \subseteq [k]} \gamma^{|M|} K_M^{(T)}} . \tag{12}$$

*Proof.* Let $\mathcal{P}(S)$ denote the power set of a given set $S$:

$$\mathcal{P}(S) := \{\forall S' : S' \subseteq S\} .$$

Let $\mathcal{P}(S, i)$ denote the subset of $\mathcal{P}(S)$ with a fixed cardinality $i \in \mathbb{N}$:

$$\mathcal{P}(S, i) := \{\forall S' : S' \subseteq S, |S'| = i\} .$$

By using this notation, we write the numerator in Equation (8):

$$\sum_{M \subseteq [k]} e^{\beta |M|} N_M^{(t)}(m) = \sum_{i=0}^{k} e^{\beta i} \sum_{M \in \mathcal{P}([k], i)} N_M^{(t)}(m)$$

$$= \sum_{i=0}^{k} e^{\beta i} \sum_{M \in \mathcal{P}([k], i)} \sum_{[k] \supseteq S \supseteq M} (-1)^{|S|-i} K_S^{(t)}(m)$$

$$= \sum_{S \subseteq [k]} K_S^{(t)}(m) \left( \sum_{i=0}^{|S|} e^{\beta i} \sum_{M \in \mathcal{P}(S, i)} (-1)^{|S|-i} \right)$$

$$= \sum_{S \subseteq [k]} K_S^{(t)}(m) \left( \sum_{i=0}^{|S|} (-1)^{|S|-i} \binom{|S|}{i} e^{\beta i} \right)$$

$$= \sum_{S \subseteq [k]} \left( e^\beta - 1 \right)^{|S|} K_S^{(t)}(m) ,$$

where the last step applies the binomial theorem; the resulting identity remains valid at $\beta = 0$ under the convention $0^0 := 1$. Now, the estimator in Equation (8) can be rewritten with $\gamma = e^\beta - 1$. □

**Corollary 4.2** (Mixtures of $n$-grams)**.** *The estimator in Equation* (8) *admits the following mixture interpretation:*

- *Chooses an order* $i \in [0, k]$ *with weights* $\{\lambda_i^{(T)}\}$,
- *Chooses a size-$i$ pattern* $M$ ($|M| = i$) *with weights* $\{\lambda_M^{(T)}\}$,
- *Predicts with the pattern model* $\hat{q}_M^{(T)}(m) = \dfrac{K_M^{(T)}(m)}{K_M^{(T)}}$.

*Proof.* The form in Lemma 4.1 is a mixture of

$$\hat{q}_i^{(T)}(m) := \frac{\displaystyle\sum_{M \in \mathcal{P}([k], i)} K_M^{(T)}(m)}{\displaystyle\sum_{M \in \mathcal{P}([k], i)} K_M^{(T)}} ,$$

---

[2]Under the convention $0^0 := 1$, Equation (12) is a polynomial identity in $\gamma$ and remains valid at $\beta = 0$ ($\gamma = 0$), where it recovers the unigram estimator $K_\emptyset^{(T)}(m) / K_\emptyset^{(T)}$.

with weights $\lambda_i^{(T)}$ which are defined as follows

$$\lambda_i^{(T)} := \frac{\gamma^i K_i^{(T)}}{\displaystyle\sum_{i=0}^{k} \gamma^i K_i^{(T)}}, \quad \text{for } K_i^{(T)} := \sum_{M \in \mathcal{P}([k],i)} K_M^{(T)}.$$

Moreover, $\hat{q}_i^{(T)}(m)$ is itself a mixture of

$$\hat{q}_M^{(T)}(m) := \frac{K_M^{(T)}(m)}{K_M^{(T)}}, \quad \text{with weights } \lambda_M^{(T)} := \frac{K_M^{(T)}}{K_i^{(T)}}.$$

$\square$

### D.3. Approximation to Optimal $\beta$.

This section approximates the optimal choice of $\beta$ in Equation (8) for Markov chains with a uniform Dirichlet prior. Recall that we have defined $\gamma = e^\beta - 1$ in Appendix D. We first derive an approximation of $\gamma$ and convert it to an approximation for $\beta$.

As explained in Section 2.2, the Bayes-optimal predictor is

$$\frac{N_{u_T}(m) + \alpha}{N_{u_T} + |V|\alpha}. \tag{16}$$

where $N_{u_T} = \sum_{j \in V} N_{u_T}(j)$ denotes the count of full query matches given context $u_T$. Consequently, Equation (16) minimizes the risk within the parametric family defined by:

$$\left\{ \frac{N_{u_T}(m) + a}{N_{u_T} + |V|a} : a > 0 \right\}.$$

Differentiating the loss with respect to the parameter $\{a\}$ and evaluating at the optimum $a = \alpha$ yields the first-order condition

$$\mathbb{E}\left[ -\frac{1}{N_{u_T}(m) + \alpha} + \frac{|V|}{N_{u_T} + |V|\alpha} \right] = 0.$$

We have shown that Equation (8) can be rewritten as Equation (12). Dividing the numerator and denominator by $\gamma^k$ and separating the exact-match term $M = [k]$ gives

$$\mathcal{T}(\boldsymbol{x})(m) = \frac{K_{[k]}^{(T)}(m) + \displaystyle\sum_{M \subset [k]} \gamma^{|M|-k} K_M^{(T)}(m)}{K_{[k]}^{(T)} + \displaystyle\sum_{M \subset [k]} \gamma^{|M|-k} K_M^{(T)}}.$$

Taking the gradient with respect to $\gamma$ and setting it to 0:

$$\mathbb{E}\left[ -\frac{\displaystyle\sum_{M \subset [k]} (|M|-k)\gamma^{|M|-k-1} K_M^{(T)}(m)}{K_{[k]}^{(T)}(m) + \displaystyle\sum_{M \subset [k]} \gamma^{|M|-k} K_M^{(T)}(m)} + \frac{\displaystyle\sum_{M \subset [k]} (|M|-k)\gamma^{|M|-k-1} K_M^{(T)}}{K_{[k]}^{(T)} + \displaystyle\sum_{M \subset [k]} \gamma^{|M|-k} K_M^{(T)}} \right] = 0. \tag{17}$$

**Taylor Expansion.** We approximately solve this equation for $\gamma$ by the following Taylor approximation.

**Lemma D.1** (First-Order Approximation for Ratios). *Let $X$ and $Y$ be random variables with means $\mu_X, \mu_Y$. The first-order Taylor approximation for the expectation of the function $f(X,Y) = \frac{X}{c+Y}$ around the point $(\mu_X, \mu_Y)$ is given by:*

$$\mathbb{E}\left[ \frac{X}{c+Y} \right] \approx \frac{\mu_X}{c + \mu_Y}. \tag{18}$$

*Proof.* Consider the function $f(x, y) = x(c + y)^{-1}$. We perform a multivariate Taylor expansion of $f(x, y)$ around the mean vector $\boldsymbol{\mu} = (\mu_X, \mu_Y)$. The first-order expansion is:

$$f(x, y) \approx f(\boldsymbol{\mu}) + (\nabla f(\boldsymbol{\mu}))^\top (\mathbf{z} - \boldsymbol{\mu}),$$

where $\mathbf{z} = (x, y)^\top$. Taking the expectation of the Taylor expansion eliminates the first-order term as $\mathbb{E}[\mathbf{z} - \boldsymbol{\mu}] = 0$. The expectation of the linear form yields the result. $\square$

Let $A^{(T)}(m), B^{(T)}(m), C^{(T)}, D^{(T)}$ denote the following random variables

$$A^{(T)}(m) := \sum_{M \subset [k]} (|M| - k)\gamma^{|M| - k - 1} K_M^{(T)}(m),$$

$$B^{(T)}(m) := \sum_{M \subset [k]} \gamma^{|M| - k} K_M^{(T)}(m),$$

$$C^{(T)} := \sum_{m \in V} A^{(T)}(m),$$

$$D^{(T)} := \sum_{m \in V} B^{(T)}(m).$$

We apply Lemma D.1 to approximate Equation (17) around the means of $A^{(T)}(m), B^{(T)}(m), C^{(T)}, D^{(T)}$ conditioned on $K_{[k]}^{(T)}(m)$ and $K_{[k]}^{(T)}$:

$$-\frac{A^{(T)}(m)}{K_{[k]}^{(T)}(m) + B^{(T)}(m)} + \frac{C^{(T)}}{K_{[k]}^{(T)} + D^{(T)}} \approx -\frac{\mathbb{E}\left[A^{(T)}(m) \mid K_{[k]}^{(T)}(m)\right]}{K_{[k]}^{(T)}(m) + \mathbb{E}\left[B^{(T)}(m) \mid K_{[k]}^{(T)}(m)\right]} + \frac{\mathbb{E}\left[C^{(T)} \mid K_{[k]}^{(T)}\right]}{K_{[k]}^{(T)} + \mathbb{E}\left[D^{(T)} \mid K_{[k]}^{(T)}\right]}.$$

Lastly, as $K_{[k]}^{(T)}(m)$ and $K_{[k]}^{(T)}$ are informative only for exact $k$-matches and $A^{(T)}(m), B^{(T)}(m), C^{(T)}, D^{(T)}$ are measuring sub-$k$-matches, we assume the following concentrations around the population means:

$$\mathbb{E}\left[A^{(T)}(m) \mid K_{[k]}^{(T)}(m)\right] \approx \mathbb{E}\left[A^{(T)}(m)\right], \quad \mathbb{E}\left[C^{(T)} \mid K_{[k]}^{(T)}\right] \approx \mathbb{E}\left[C^{(T)}\right],$$

$$\mathbb{E}\left[B^{(T)}(m) \mid K_{[k]}^{(T)}(m)\right] \approx \mathbb{E}\left[B^{(T)}(m)\right], \quad \mathbb{E}\left[D^{(T)} \mid K_{[k]}^{(T)}\right] \approx \mathbb{E}\left[D^{(T)}\right].$$

To approximate the optimal value of $\gamma$, we solve the following equation:

$$\mathbb{E}\left[-\frac{\mathbb{E}\left[A^{(T)}(m)\right]}{K_{[k]}^{(T)}(m) + \mathbb{E}\left[B^{(T)}(m)\right]} + \frac{\mathbb{E}\left[C^{(T)}\right]}{K_{[k]}^{(T)} + \mathbb{E}\left[D^{(T)}\right]}\right] = 0. \tag{19}$$

Observe that $K_{[k]}^{(T)}(m)$ is precisely $N_{u_T}(m)$ and $K_{[k]}^{(T)}$ is equal to $\sum_{j \in V} N_{u_T}(j)$. Consequently, it is sufficient to satisfy the following identities for an arbitrary constant $c$:

$$\mathbb{E}\left[A^{(T)}(m)\right] = c, \quad \mathbb{E}\left[B^{(T)}(m)\right] = \alpha, \quad \mathbb{E}\left[C^{(T)}\right] = c|V|, \quad \mathbb{E}\left[D^{(T)}\right] = \alpha|V|. \tag{20}$$

Given the symmetry of the prior and the model, the latter two conditions are satisfied provided the first two hold, and vice versa. As $c$ is arbitrary, we just need to verify $\mathbb{E}\left[B^{(T)}(m)\right] = \alpha$ or $\mathbb{E}\left[D^{(T)}\right] = \alpha|V|$.

**First-order $k = 1$.** Unfortunately, $\mathbb{E}\left[K_M^{(T)}(m)\right]$ is not tractable for high-order Markov chains. For $k = 1$, we can solve Equation (19) as follows. As the prior is uniform over all elements in $V$, we have[3]

$$\mathbb{E}\left[K_\emptyset^{(T)}(m)\right] = \frac{T - k - 1}{|V|},$$

---

[3]We count the $T - k - 1$ candidate positions $s \in [k', T - 1]$ that strictly precede the query, each contributing $\Pr(x_s = m) = 1/|V|$; this is the candidate range of Definition 4.1 excluding the query position $T$ itself.

which leads to the following:

$$\mathbb{E}\left[B^{(T)}(m)\right] = \gamma^{-1}\frac{T-k-1}{|V|}\,.$$

This is satisfied by

$$\gamma = \frac{T-k-1}{\alpha|V|}\,, \quad \text{or} \quad \beta = \ln\left(1+\frac{(T-k-1)}{\alpha|V|}\right)\,.$$

**Higher-orders** $k > 1$. For higher-orders, we note that

$$\mathbb{E}\left[D^{(T)}+K_{[k]}^{(T)}\right] = \gamma^{-k}\mathbb{E}\left[Z(\gamma)\right]\,, \quad \text{for} \quad Z(\gamma) = \sum_{M\subseteq[k]}\gamma^{|M|}K_M^{(T)}\,.$$

$Z(\gamma)$ is exactly the normalization constant in Equation (8). To satisfy Equation (20), we need

$$\gamma^{-k}\mathbb{E}\left[Z(\gamma)\right] = \left(\mathbb{E}\left[K_{[k]}^{(T)}\right]+\alpha|V|\right)\,. \tag{21}$$

The right-hand side is independent on the choice of $\gamma$ and the left-hand side is monotonically decreasing in $\gamma$. Therefore, the value that solves Equation (20) can be found empirically in two ways. First, it is possible to identify the value of $\gamma$ by a simple line search. Alternatively, it is possible to estimate $\mathbb{E}\left[K_M^{(T)}\right]$ and solve the roots of the polynomial in Equation (21).

To get a closed-form, we use the following approximation:

$$\mathbb{E}\left[K_M^{(T)}\right] \approx \frac{T-k-1}{|V|^{|M|}}\,. \tag{22}$$

This is derived by assuming that each length-$k$ context has the same marginal probability and the contexts at different positions are independent of each other. Then, Equation (21) yields

$$\gamma^{-k} \approx \frac{\dfrac{T-k-1}{|V|^k}+\alpha|V|}{(T-k-1)\left(\dfrac{\gamma}{|V|}+1\right)^k}\,.$$

Solving for $\gamma$, we obtain:

$$\gamma \approx \frac{|V|}{\sqrt[k]{1+\dfrac{\alpha|V|^{k+1}}{T-k-1}}-1}\,.$$

Note that these approximations are valid when the random variables $A^{(T)}(m), B^{(T)}(m), C^{(T)}, D^{(T)}$ have well-concentrated around their mean and the second-order error term omitted in Equation (18) is small. Moreover, for higher-order approximations, Equation (22) is a coarse approximation.

**Pseudo-count view.** Our approach above is equivalent to assuming that $\tilde{\alpha}_m^{(T)}$ concentrates around its expected value, $f(\beta) \coloneqq \mathbb{E}\left[\tilde{\alpha}_m^{(T)}\right]$ and approximating the function $f$ by simplifying the distributional properties of the data. That is, we have chosen $\beta$ such that $\mathbb{E}\left[\tilde{\alpha}_m^{(T)}\right] \approx \alpha$. Thus, we expect our approach to work well whenever the sequences are long enough so that Equation (14) matches Equation (7).

## E. Label-Permutation Symmetry

This appendix observes a symmetry property of the in-context Markov chain tasks in Section 2.2 and explains its implication for disentangled transformers trained by gradient flow. For label-symmetric tasks and symmetric initialization, gradient flow preserves label-permutation invariance. In the two-layer copy-and-match setup, this invariance constrains the content-based comparisons to equality tests between copied tokens, matching the basic comparison used by the soft context-matching construction in Proposition 3.1. Thus, the result characterizes the constraint imposed by symmetry, but does not identify which symmetry-compatible parameters are selected by training or optimal for prediction.

## E.1. Definitions

Let $\mathbb{P}$ be a distribution over sequences $x \in V^T$.

**Definition E.1** (Label symmetry). We say that $\mathbb{P}$ has *label symmetry* if for any permutation $\sigma : V \to V$,

$$\mathbb{P}(x_1, \ldots, x_T) = \mathbb{P}(\sigma(x_1), \ldots, \sigma(x_T)). \tag{23}$$

Label symmetry states that the distribution depends on the equality pattern among tokens rather than on the token identities themselves.

**Definition E.2** (Label-permutation invariant predictor). For a predictor $\mathcal{T}$ and a permutation $\sigma : V \to V$, define

$$\mathcal{T}^{(\sigma)}(x_1, \ldots, x_T) := \sigma^{-1}\left(\mathcal{T}(\sigma(x_1), \ldots, \sigma(x_T))\right). \tag{24}$$

We say that $\mathcal{T}$ is *label-permutation invariant* if $\mathcal{T}^{(\sigma)} = \mathcal{T}$ for every permutation $\sigma : V \to V$. We denote this class by

$$\mathcal{M} := \left\{ \mathcal{T} : \mathcal{T}^{(\sigma)} = \mathcal{T}, \ \forall \sigma : V \to V \right\}.$$

## E.2. Markov Chains Have Label Symmetry

Let $\boldsymbol{\pi}$ be a kernel for an order-$k$ Markov chain and let $\sigma : V \to V$ be a permutation. Define the permuted kernel $\boldsymbol{\pi}^{(\sigma)}$ by

$$\boldsymbol{\pi}^{(\sigma)}_{(c_1, \ldots, c_k)}(m) := \boldsymbol{\pi}_{(\sigma(c_1), \ldots, \sigma(c_k))}(\sigma(m)), \quad \forall (c_1, \ldots, c_k) \in V^k, \ m \in V.$$

Given a set $S$ of Markov kernels, write $S^{(\sigma)}$ for the corresponding set of permuted kernels. Assume that the prior over kernels is invariant under label permutations:

$$\mathbb{P}(\boldsymbol{\pi} \in S) = \mathbb{P}\left(\boldsymbol{\pi} \in S^{(\sigma)}\right) \quad \text{for all measurable } S. \tag{25}$$

Assume also that the initial context distribution $\rho$ over $x_{1:k}$ is label-symmetric:

$$\rho(x_1, \ldots, x_k) = \rho(\sigma(x_1), \ldots, \sigma(x_k)). \tag{26}$$

This holds for the uniform initialization used in Section 2.2. Then the induced sequence distribution satisfies label symmetry:

$$\begin{aligned}
\mathbb{P}(x_1, \ldots, x_T) &= \mathbb{E}_{\boldsymbol{\pi}} \mathbb{P}(x_1, \ldots, x_T \mid \boldsymbol{\pi}) \\
&= \mathbb{E}_{\boldsymbol{\pi}} \mathbb{P}\left(x_1, \ldots, x_T \mid \boldsymbol{\pi}^{(\sigma)}\right) \\
&= \mathbb{E}_{\boldsymbol{\pi}} \mathbb{P}(\sigma(x_1), \ldots, \sigma(x_T) \mid \boldsymbol{\pi}) \\
&= \mathbb{P}(\sigma(x_1), \ldots, \sigma(x_T)).
\end{aligned}$$

The first equality marginalizes over the sampled transition kernel. The second equality uses the prior symmetry in Equation (25): averaging over $\boldsymbol{\pi}$ is the same as averaging over $\boldsymbol{\pi}^{(\sigma)}$. For the third equality, expanding the conditional probability gives:

$$\mathbb{P}(x_1, \ldots, x_T \mid \boldsymbol{\pi}^{(\sigma)}) = \rho(x_1, \ldots, x_k) \prod_{t=k+1}^{T} \boldsymbol{\pi}^{(\sigma)}_{(x_{t-k}, \ldots, x_{t-1})}(x_t).$$

Using Equation (26) and the definition of $\boldsymbol{\pi}^{(\sigma)}$, this becomes

$$\rho(\sigma(x_1), \ldots, \sigma(x_k)) \prod_{t=k+1}^{T} \boldsymbol{\pi}_{(\sigma(x_{t-k}), \ldots, \sigma(x_{t-1}))}(\sigma(x_t)) = \mathbb{P}(\sigma(x_1), \ldots, \sigma(x_T) \mid \boldsymbol{\pi}).$$

The final equality marginalizes over $\boldsymbol{\pi}$ again.

The independent Dirichlet prior with symmetric concentration $\text{Dirichlet}(\alpha, \ldots, \alpha)$ satisfies Equation (25). Indeed, its density

$$f(\mathbf{p}) = \frac{1}{Z(\alpha)} \prod_{i=1}^{|V|} p_i^{\alpha - 1}$$

is invariant under coordinate permutations. The joint density of an independently sampled Markov kernel is a product of such row densities, and a label permutation only permutes the coordinates within each row and the rows within the context space $V^k$. The hierarchical Dirichlet prior in Section 2.2 satisfies the same condition when each level uses symmetric concentration parameters: the base distribution is permutation invariant, and the parent-child relation between contexts is preserved by relabeling.

### E.3. Gradient Flow Preserves Label-Permutation Invariance

We first define the corresponding relabeling operation on the parameters of the disentangled transformer in Section 2.1. Let $P_\sigma$ be the permutation matrix associated with $\sigma$. Since the residual stream is a concatenation of $|V|$-dimensional token blocks, let $B_l^{(\sigma)}$ denote the block-diagonal matrix that applies $P_\sigma$ to each token block in $\boldsymbol{h}^{(l)}$. For parameters $\theta$, define $\theta^{(\sigma)}$ by

$$\left(\boldsymbol{W}_A^{(l,h)}\right)^{(\sigma)} = \left(B_{l-1}^{(\sigma)}\right)^\top \boldsymbol{W}_A^{(l,h)} B_{l-1}^{(\sigma)}, \qquad \boldsymbol{W}_O^{(\sigma)} = P_\sigma^\top \boldsymbol{W}_O B_L^{(\sigma)}.$$

The relative positional encodings are left unchanged, since they are indexed by relative position rather than by token label. With this definition, relabeling the parameters implements the relabeled predictor:

$$\mathcal{T}_{\theta^{(\sigma)}} = \mathcal{T}_\theta^{(\sigma)}. \tag{27}$$

Indeed, if $\boldsymbol{h}_{t,\theta}^{(l)}(\sigma(\boldsymbol{x}))$ is the representation produced by $\theta$ on the relabeled input, then the representation produced by $\theta^{(\sigma)}$ on $\boldsymbol{x}$ is $\left(B_l^{(\sigma)}\right)^\top \boldsymbol{h}_{t,\theta}^{(l)}(\sigma(\boldsymbol{x}))$. This holds at the input layer by the definition of one-hot relabeling. If it holds at layer $l-1$, the content-based attention scores agree because

$$\left(\left(B_{l-1}^{(\sigma)}\right)^\top q\right)^\top \left(B_{l-1}^{(\sigma)}\right)^\top \boldsymbol{W}_A^{(l,h)} B_{l-1}^{(\sigma)} \left(\left(B_{l-1}^{(\sigma)}\right)^\top k\right) = q^\top \boldsymbol{W}_A^{(l,h)} k,$$

and the RPE terms are unchanged. The values are the previous-layer representations themselves, so the same block-wise relabeling passes through the attention-weighted sum and the residual update. Finally, the definition of $\boldsymbol{W}_O^{(\sigma)}$ maps the final representation to $P_\sigma^\top$ times the original logits on the relabeled input, which is exactly the output relabeling in Equation (24).

**Theorem E.1** (Label-permutation invariance). *Let $\mathbb{P}$ be any distribution with label symmetry. Let $\mathcal{T}_{\theta(0)}$ be a disentangled transformer whose initialization is fixed by every parameter relabeling above: $\theta(0)^{(\sigma)} = \theta(0)$ for all permutations $\sigma : V \to V$. Equivalently, the initialization satisfies*

$$\boldsymbol{W}_A^{(l,h)} = \left(B_{l-1}^{(\sigma)}\right)^\top \boldsymbol{W}_A^{(l,h)} B_{l-1}^{(\sigma)}, \quad \text{for all } l, h, \qquad \boldsymbol{W}_O = P_\sigma^\top \boldsymbol{W}_O B_L^{(\sigma)},$$

*for every $\sigma$, with the RPE parameters unrestricted. A simple sufficient initialization satisfying both conditions for every label permutation is to set $\boldsymbol{W}_A^{(l,h)}(0) = \boldsymbol{0}$ for every layer $l$ and head $h$, and to set $\boldsymbol{W}_O(0) = \boldsymbol{0}$. Then the gradient flow $\theta(r)$ induced by the population next-token prediction loss satisfies*

$$\mathcal{T}_{\theta(r)} \in \mathcal{M}, \quad \text{for all } r > 0.$$

*Proof.* Let $\mathcal{L}$ be the population next-token prediction loss:

$$\mathcal{L}(\theta) = \mathbb{E}_{\boldsymbol{x}\sim\mathbb{P}} \left[ -\log \mathcal{T}_\theta(x_T \mid \boldsymbol{x}_1^{T-1}) \right].$$

By label symmetry,

$$\mathcal{L}(\theta) = \mathbb{E}_{\boldsymbol{x}\sim\mathbb{P}} \left[ -\log \mathcal{T}_\theta(\sigma(x_T) \mid \sigma(\boldsymbol{x}_1^{T-1})) \right]$$
$$= \mathbb{E}_{\boldsymbol{x}\sim\mathbb{P}} \left[ -\log \mathcal{T}_{\theta^{(\sigma)}}(x_T \mid \boldsymbol{x}_1^{T-1}) \right]$$
$$= \mathcal{L}(\theta^{(\sigma)}).$$

Let $T_\sigma$ denote the linear map $\theta \mapsto \theta^{(\sigma)}$ on the parameter vector. The map $T_\sigma$ is orthogonal, because it acts by left and right multiplication by permutation matrices on $\boldsymbol{W}_A$ and $\boldsymbol{W}_O$, and leaves the RPE parameters unchanged. The identity above says that $\mathcal{L}(\theta) = \mathcal{L}(T_\sigma \theta)$. To differentiate this identity explicitly, fix an arbitrary perturbation $u$ and consider the one-dimensional path $\theta_0 + \epsilon u$. Since

$$\mathcal{L}(\theta_0 + \epsilon u) = \mathcal{L}\big(T_\sigma(\theta_0 + \epsilon u)\big) = \mathcal{L}(T_\sigma \theta_0 + \epsilon T_\sigma u),$$

differentiating with respect to $\epsilon$ at $\epsilon = 0$ gives

$$\langle \nabla \mathcal{L}(\theta_0), u \rangle = \langle \nabla \mathcal{L}(T_\sigma \theta_0), T_\sigma u \rangle = \langle T_\sigma^\top \nabla \mathcal{L}(T_\sigma \theta_0), u \rangle \,.$$

Because this holds for every $u$,

$$\nabla \mathcal{L}(\theta_0) = T_\sigma^\top \nabla \mathcal{L}(T_\sigma \theta_0) \,.$$

Multiplying by $T_\sigma$ and using $T_\sigma \theta_0 = \theta_0^{(\sigma)}$, we obtain

$$T_\sigma \nabla \mathcal{L}(\theta_0) = \nabla \mathcal{L}(\theta_0^{(\sigma)}) \,,$$

which is the equivariance of the full gradient.

Now suppose $T_\sigma \theta_0 = \theta_0$. Then the equivariance relation implies

$$T_\sigma \big( -\nabla \mathcal{L}(\theta_0) \big) = -\nabla \mathcal{L}(T_\sigma \theta_0) = -\nabla \mathcal{L}(\theta_0) \,.$$

Thus the gradient-flow vector field is tangent to the fixed-point set $\{\theta : T_\sigma \theta = \theta\}$. Since the initialization satisfies $T_\sigma \theta(0) = \theta(0)$ for every $\sigma$, gradient flow remains in this fixed-point set for every $r > 0$; the argument applies to every $\sigma$. Therefore $\theta(r)^{(\sigma)} = \theta(r)$ for every $\sigma$. Using Equation (27),

$$\mathcal{T}_{\theta(r)}^{(\sigma)} = \mathcal{T}_{\theta(r)^{(\sigma)}} = \mathcal{T}_{\theta(r)} \,,$$

so $\mathcal{T}_{\theta(r)} \in \mathcal{M}$ for all $r > 0$. $\qquad\square$

### E.4. Consequence for Two-Layer Context Matching

We use one elementary fact about permutation-invariant blocks.

**Lemma E.1.** *If $A \in \mathbb{R}^{|V| \times |V|}$ satisfies*

$$P_\sigma A P_\sigma^\top = A \,, \quad \forall \sigma : V \to V,$$

*then there exist scalars $c_{\mathrm{diag}}, c_{\mathrm{all}} \in \mathbb{R}$ such that*

$$A = c_{\mathrm{diag}} \boldsymbol{I}_{|V| \times |V|} + c_{\mathrm{all}} \boldsymbol{1}_{|V|} \boldsymbol{1}_{|V|}^\top. \tag{28}$$

*Proof.* First take any two indices $i, j \in V$, and let $\sigma$ be the transposition that swaps $i$ and $j$. Since $P_\sigma A P_\sigma^\top = A$, the diagonal entries $A_{ii}$ and $A_{jj}$ must be equal. Thus all diagonal entries share a common value. Next take any two ordered pairs $(i, j)$ and $(i', j')$ with $i \neq j$ and $i' \neq j'$. There is a permutation sending $i$ to $i'$ and $j$ to $j'$, so invariance gives $A_{ij} = A_{i'j'}$. Thus all off-diagonal entries share a common value. Writing the common off-diagonal value as $c_{\mathrm{all}}$ and the difference between the common diagonal and off-diagonal values as $c_{\mathrm{diag}}$ gives Equation (28). $\qquad\square$

**Corollary E.1** (Generalized context matching under label symmetry)**.** *Consider a two-layer disentangled transformer satisfying the assumptions of Theorem E.1. Suppose the first layer implements the copying heads from Section 3.2.1 throughout training, and the second-layer content-based attention matrix is trained by gradient flow on a label-symmetric task. Then each $|V| \times |V|$ token-comparison block of the second-layer content-based attention matrix remains equivalent, up to an input-independent additive attention shift, to a scalar multiple of the identity. Consequently, if the value and output maps aggregate successor tokens as in Proposition 3.1, the resulting predictor has the generalized soft context-matching form*

$$\mathcal{T}(\boldsymbol{x})(m) = \frac{\displaystyle\sum_{M \subseteq [k] \times [k]} e^{|\boldsymbol{\beta}|_M} N_M^{(T)}(m)}{\displaystyle\sum_{M \subseteq [k] \times [k]} e^{|\boldsymbol{\beta}|_M} N_M^{(T)}} \,,$$

*where*

$$M_s^{(t)} := \big\{ (r_1, r_2) \in [k] \times [k] : x_{t - r_1} = x_{s - r_2} \big\} \subseteq [k] \times [k],$$

$N_M^{(T)}(m)$ *counts previous positions $s$ with $M_s^{(T)} = M$ and successor token $x_s = m$, $N_M^{(T)} = \sum_{m \in V} N_M^{(T)}(m)$, and*

$$|\boldsymbol{\beta}|_M := \sum_{(a,b) \in M} \beta_{a,b}.$$

*Proof.* By Theorem E.1, the second-layer content-based attention matrix remains invariant under simultaneous permutation of token labels. In the copied representation, this permutation acts separately on each $|V|$-dimensional token block, so each $|V| \times |V|$ token-comparison block is permutation invariant. By Lemma E.1, each block has the form

$$c_{\text{diag}} \boldsymbol{I}_{|V| \times |V|} + c_{\text{all}} \mathbf{1}_{|V|} \mathbf{1}_{|V|}^\top.$$

In the copied representation from Section 3.2.1, each token block is one-hot and has $L_1$ norm 1. The $\mathbf{1}_{|V|} \mathbf{1}_{|V|}^\top$ term therefore adds the same constant to the attention score for every candidate position $s$, for a fixed query and block. This additive shift is removed by the softmax normalization, so each block is equivalent, for attention weights, to a scalar multiple of the identity.

Let $\beta_{r_1, r_2}$ be the scalar associated with the block comparing query lag $r_1$ to key lag $r_2$. The contribution of this block to the second-layer attention score is $\beta_{r_1, r_2}$ exactly when the copied query token at lag $r_1$ equals the copied key token at lag $r_2$, and is 0 otherwise. Summing over all block pairs, the context-matching score is therefore proportional to

$$\sum_{(r_1, r_2) \in [k] \times [k]} \beta_{r_1, r_2} \mathbb{I}\{x_{t-r_1} = x_{s-r_2}\}.$$

Grouping candidate positions $s$ by the induced match mask $M_s^{(T)}$ gives attention weights proportional to $e^{|\boldsymbol{\beta}|_M}$. If the value and output maps aggregate successor-token one-hot vectors as in Proposition 3.1, normalizing these weighted mask-conditioned transition counts gives the displayed generalized soft context-matching estimator. $\square$

The corollary generalizes the aligned match mask in Definition 3.1: instead of comparing lag $r$ only with lag $r$, the most general symmetry-compatible second-layer comparison may assign separate weights to pairs of lags $(r_1, r_2)$. This should not be interpreted as saying that every trained transformer must use this estimator. It says that, under the stated architecture and gradient-flow idealization, label symmetry constrains the content-based attention computation to compare token equality rather than token identities. Heuristically, for the Markov prediction task, we expect the useful parameters to concentrate on the offset-aligned comparisons used in Proposition 3.1: these are the super-diagonal blocks in the construction's block notation, which compare the query context to the predecessor context of each candidate successor token. This provides intuition for why the main-text construction and the trained models in Figure 3 exhibit identity-based, offset-aligned comparison structure.

