# OpenReview forum: "Induction Heads Interpolate N-Grams"
_ICML.cc/2026/Conference — ICML 2026 regular_

### Official Review · Reviewer_TcBr · 2026-03-12

**Soundness:** 3
**Presentation:** 3
**Significance:** 3
**Originality:** 3
**Overall Recommendation:** 5
**Confidence:** 2

**Summary:**

This paper theoretically advances the study of induction heads. The analysis is predicated on a simplified two-layer disentangled transformer model, which is trained to perform in-context prediction on sequences generated from Markov models.
Theoretically, the authors demonstrate that by carefully configuring the transformer's heads and attention weights, the model can compute a score that is geometrically weighted by the number of matching tokens (ranging from 0 to $k$) within a $k$-gram pattern. Building upon this, the paper shows that the weight assigned to the BOS (Beginning-Of-Sequence) token elegantly corresponds to add-$\alpha$ smoothing in standard $n$-gram models. Additionally, it establishes that the mechanism can realize an exponential Hamming kernel, functionally similar to a Nadaraya-Watson estimator.
Furthermore, through empirical experiments that manipulate the softmax temperature, the authors indirectly validate that trained transformers practically execute this exact type of statistical smoothing. The profound impact of the BOS token on mitigating sparsity and improving performance is particularly striking and well-supported by the results.

**Compliance With Llm Reviewing Policy:**

Affirmed.

**Final Justification:**

**Reasons for changing the score to Accept:**
I am raising my score from Weak Accept to Accept for the following two reasons:
* **Direct mechanistic evidence:** The authors successfully addressed my primary concern by pointing out the internal weight analysis in the Appendix and committing to move this discussion to the main text.
* **Relationship with Kneser-Ney:** They provided a clear and constructive explanation regarding the connection to Kneser-Ney smoothing, which they also promised to incorporate into the revised paper.

**Key Questions For Authors:**

# Questions

* The paper elegantly connects the transformer's behavior to Jelinek-Mercer-style interpolation. How do the authors view the relationship between the proposed mechanism and Kneser-Ney smoothing? I would appreciate your insights on this connection.

* In the current theoretical framework, it appears that the matching score depends solely on the number of matched tokens within the $n$-gram, rather than on specific matching patterns. Do the authors believe that this limitation in expressive power could be resolved by other mechanisms (e.g., by adding more layers)?

**Limitations:**

yes

**Strengths And Weaknesses:**

### Strengths

**Soundness:**
The theoretical proofs are rigorous and sound. The paper clearly delineates the boundaries of its construction, explicitly acknowledging that the score depends solely on the number of matching tokens in the $n$-gram rather than on specific token combinations. Furthermore, the connections drawn to multiple $n$-gram smoothing techniques are mathematically well-founded. The experimental setup is also straightforward, appropriate, and clear.

**Presentation:**
The paper is well-organized and clearly written. The construction of the disentangled transformer model, which serves as the foundation for the theorems and constructive proofs, is lucidly presented. In particular, Figure 1 effectively illustrates the core mechanism of the model in an intuitive manner.

**Significance:**
The constructive proof of the model offers deep insights into the types of statistical smoothing mechanisms that can practically exist within attention heads, making a solid theoretical contribution to the understanding of transformer learning dynamics. Furthermore, the empirical observation that the BOS token behaves as a pseudo-count—exactly as the theory predicts—is a remarkable and impactful finding.

**Originality:**
The originality of this work lies in providing a theoretical demonstration that specific types of $n$-gram smoothing can be inherently represented within the transformer architecture. Combining this theoretical framework with empirical results, which suggest that such smoothing operations are actually occurring in trained models, highlights a novel and original perspective on the mechanisms of in-context learning.

### Weaknesses

**Soundness:**
While the experimental results—manipulating the attention temperature and the presence of the BOS token—strongly suggest that the theoretically proposed smoothing mechanisms are indeed occurring, the evidence remains indirect. The empirical findings do not immediately or directly prove that the exact internal representations and mechanisms derived in the theory are actually formed within the trained transformer. This lack of direct mechanistic evidence (e.g., probing internal activations or attention heads to perfectly match the constructed weights) slightly weakens the overall soundness of the empirical claims.

**Presentation:**
No major issues were found. The presentation is clear and adequate.

**Significance:**
The theoretical analysis and empirical validation rely heavily on a simplified "disentangled" two-layer transformer model, which structurally differs from typical transformers used in practice (i.e., lacking MLPs and standard additive residual connections). Consequently, it remains unclear whether the same smoothing mechanisms and theoretical guarantees would hold—both theoretically and empirically—in standard, deeper transformer architectures. This gap limits the immediate broader significance of the findings to practical LLM settings.

**Originality:**
No major issues were found. The theoretical connection between attention mechanisms and classical smoothing remains highly original.

---

> ### Author Rebuttal · Authors · 2026-03-31
>
> We thank the reviewer for the careful reading and positive evaluation. We provide supplementary material with all new experiments at [this anonymous link](https://anonymous.4open.science/r/icml_2026_rebuttal_supplementary-BCD5/main.pdf).
>
> **Indirect vs direct mechanistic evidence.** The reviewer raises an excellent point. While the main text highlights behavioral interventions (attention temperature and BOS token), Appendix B provides the direct mechanistic probing requested by the reviewer. Sections B.3 and B.4 (Figures 26 and 34) compare the internal weights, RPE patterns, and $W_A$ matrices of trained transformers directly with our theoretical construction. The trained networks form representations matching the theory, modulo invariant weight transformations (as is standard when comparing learned vs. constructed networks in deep learning). We agree that this was not sufficiently illustrated in the main text, we will add a paragraph to the main experiments section summarizing this direct internal evidence.
>
> **Disentangled vs standard architectures (new experiments).** We view our work as foundational: because induction heads are fundamental computational primitives originally discovered in two-layer attention-only models (Elhage et al., 2021), it is standard practice in the ICL theory literature to analyze them in simplified settings (Von Oswald et al., 2023; Nichani et al., 2024). We note that our construction extends naturally to standard architectures: with learned embeddings of sufficient dimension, the value matrices can project tokens into distinct subspaces of the residual stream, implementing the same lag-specific copying that one-hot concatenation provides explicitly. The disentangled architecture was chosen for interpretability, not out of necessity. To verify this empirically, we have added experiments with **standard transformers** (learned token embeddings with $d_{\text{model}}=64$, additive residual connections, LayerNorm, MLP layers, and learned relative positional biases) on the hierarchical Dirichlet task. These models achieve the same KL performance and exhibit the same two-stage mechanism — Layer 1 copy heads and Layer 2 partial-match interpolation — confirming that the smoothing behavior is a general property of the attention computation (supplement, Figure 1 for KL comparison; Figure 3 for induction head analysis; Figures 4–5 for attention patterns). Validating these findings in large pretrained LLMs on text data is exciting future work but beyond the scope of this theoretical contribution.
>
> **Connection to Kneser-Ney smoothing.** We thank the reviewer for this insightful question. Our estimator shares the philosophy of interpolating between higher- and lower-order contexts but diverges from Kneser-Ney in three key ways:
>
> 1. *Gapped matches*: KN relies on strict recursive back-off through contiguous suffixes. Our estimator sums over all subsets $M \subseteq [k]$, including non-contiguous ("gapped") matches that KN would discard entirely.
> 2. *Soft vs absolute discounting*: KN subtracts a fixed constant from observed counts. Our mechanism uses a softmax temperature $\beta$ to create multiplicative, data-dependent pseudo-counts $\tilde{\alpha}_m^{(T)}(\gamma)$ that scale smoothly with context similarity.
> 3. *Raw frequencies vs continuation probabilities*: KN's defining feature is backing off to the number of unique context *types* that precede a word. Our transformer interpolates over raw frequencies of partial matches $K_M^{(T)}(m)$, not type-based counts.
>
> In summary, KN achieves smoothing via rigid discounting and type-based continuation, while the transformer approximates a related solution through soft, data-dependent, non-contiguous similarity matching. We will add a paragraph discussing this connection in the revised paper.
>
> **Pattern-dependent matching scores.** In the basic construction (Proposition 3.1), the weight $ e^{\lvert \beta \rvert_M} $ depends only on $|M|$ (how many positions match), not on which specific positions match. However, Corollary 5.2 shows that, under label symmetry, a more general form with independent $\beta_{r_1, r_2}$ for each position pair is learnable. In this case, matching at position $r_1 = 1$ (recent context) can receive a different weight from a match at $r_2 = 2$ (older context), so the matching pattern *does* matter. Additional layers could enable even richer pattern dependence, e.g., by conditioning interpolation weights on local count statistics. This is an interesting direction for future theoretical work.

---

> > ### Author Rebuttal · Reviewer_TcBr · 2026-04-02
> >
> > I would like to thank the authors for their highly constructive rebuttal.
> >
> > Regarding my primary concern about the need for direct mechanistic evidence (i.e., internal weights) rather than solely relying on indirect behavioral validation, I appreciate that you have provided positive results in the Appendix. Moving this discussion to the main text will clearly strengthen the paper. Furthermore, thank you for the detailed and convincing explanation of the relationship with Kneser-Ney smoothing. I completely agree with your points, and adding this discussion to the revised version will be highly beneficial. Since my main concerns and questions have been thoroughly addressed, I am raising my score from Weak Accept to Accept.
> >
> > To clarify my previous point regarding the matching score: my intention was to highlight the inherent constraint that the assigned weight scales linearly with the number of matched tokens *for each pattern*. While the mechanism evaluates patterns, this strict linear dependency on the count might not be sufficiently expressive for all language modeling scenarios, meaning it may not be a universal solution (which is perhaps expected, but worth noting).

---

> > > ### Author Response · Authors · 2026-04-08
> > >
> > > Thank you for the careful reading and very positive assessment, and especially for engaging so thoughtfully with both the mechanistic and statistical perspectives of the paper. We are very encouraged that the rebuttal resolved your main concerns, and in the revision we will make sure to move the direct mechanistic evidence into the main text. Regarding your point about the expressivity of the matching score, we think this is a very helpful clarification, and we agree it is worth stating more explicitly in the paper. While our theory identifies a broad and interpretable family of soft context-matching estimators, the goal of the paper is not to claim a universal solution to language modeling; rather, our contribution is to show that transformers can naturally realize this specific, statistically meaningful smoothing mechanism for in-context learning.

---

### Official Review · Reviewer_PJrz · 2026-03-12

**Soundness:** 3
**Presentation:** 3
**Significance:** 2
**Originality:** 2
**Overall Recommendation:** 5
**Confidence:** 3

**Summary:**

The subject of this work is to study the estimator implemented by induction heads in transformers performing in-context learning. The authors consider a setting where transformers are trained on sequences generated by order-k Markov chains and try to determine the computation implemented by the attention mechanism.
The main idea of the paper is to show that induction heads implement a soft context matching estimator which aggregates statistics from partial context matches. The attention mechanism gives exponentially larger weight to context with higher overlap with the current context.
The authors provide a proof that a two-layer disentangled transformer can implement this estimator. The paper also shows that the use of a BOS token produces counts which correspond to a Dirichlet smoothing, while a finite attention temperature produces interpolation across different context orders similar to Jelinek-Mercer smoothing.
Finally, the authors run experiments on transformers trained on synthetic sequences generated from Markov chains and compare the predictions of the transformer with the Bayes-optimal estimator.

**Compliance With Llm Reviewing Policy:**

Affirmed.

**Final Justification:**

The authors have thoroughly addressed the points raised in my review (except for the fact that the experiments are always on order-2 Markov chains with vocabulary size n = 5 which is not very realistic). I support publishing this paper provided the authors add these new experiments they have included in their rebuttal to the paper.

**Key Questions For Authors:**

- To what extent does the estimator characterized in the paper depend on the specific disentangled transformer architecture? Would the same smoothing behavior appear in standard transformers with residual additions and MLP layers?

**Limitations:**

Yes.

**Strengths And Weaknesses:**

Strengths
- The connection between induction heads and classical smoothing techniques from statistics is interesting. In particular, the interpretation of attention as an interpolation across partial context matches shows a connection between transformers and n-gram models.
- The proof showing that a two-layer transformer can implement the proposed estimator provides a clear interpretation of the role of the attention heads.
- The analysis highlights an interesting role of the BOS token, which acts as a pseudo-count mechanism similar to Dirichlet smoothing.


Weaknesses
- The scope of the theoretical result is narrow. The proof relies on a very specific architecture (disentangled transformers without MLPs and with concatenated residual streams). It would be good to see if the conclusions apply to standard transformer architectures used in modern language models.
- The experimental section is limited. The experiments are performed on synthetic Markov chains with a vocabulary of size 5 and order 2, which is far from realistic language modeling settings. The results confirm the theoretical construction, but they do not provide evidence that the mechanism described in the paper appears in practical models.
- The novelty of the contribution is a little limited. Previous work has already shown that induction heads implement context matching mechanisms and that transformers can behave as statistical estimators in in-context learning settings. The current paper mainly refines this understanding by providing a more precise characterization of the estimator.
- The empirical evaluation focuses almost entirely on reproducing the theoretical construction. As a result, it is not clear whether the smoothing behavior described in the paper is actually learned by transformers in realistic training settings.
- On a different note, the writing of the paper is very clear but it lacks some transitions between sections / indications of where the paper is headed. Some equations/results are introduced without much justification. Adding a few extra explanations would render the presentation excellent.

---

> ### Author Rebuttal · Authors · 2026-03-31
>
> We thank the reviewer for the constructive feedback. We provide supplementary material with all new experiments at [this anonymous link](https://anonymous.4open.science/r/icml_2026_rebuttal_supplementary-BCD5/main.pdf).
>
> **Narrow scope / standard transformers (new experiments).**
> We note that our constructive proof extends naturally to standard architectures: with learned embeddings of sufficient dimension, the value matrices can project tokens into distinct subspaces of the residual stream, implementing the same lag-specific copying that one-hot concatenation provides explicitly. The disentangled architecture was chosen because it makes the weight matrices directly interpretable, not because the mechanism requires it.
>
> To verify this empirically, we have added experiments with **standard transformers** (learned token embeddings with $d_{\text{model}}=64$, additive residual connections, LayerNorm, MLP layers, and learned relative positional biases) trained on the hierarchical Dirichlet task — removing all architectural specificities of the disentangled construction. These models achieve the same KL performance as the disentangled construction and the fully trained disentangled transformer, substantially outperforming all add-$\alpha$ baselines. Mechanistic analysis confirms the same two-stage mechanism: Layer 1 heads learn sharp copy patterns at specific lags, and the Layer 2 head implements context matching with non-trivial attention on partial matches (supplement, Figure 1 for KL comparison; Figures 3–5 for mechanistic evidence). This demonstrates that the interpolation mechanism is a general property of the attention computation, not an artifact of the disentangled architecture.
>
> We note that using simplified architectures and setups for theoretical analysis is standard practice in the ICL literature (Elhage et al., 2021; Von Oswald et al., 2023; Nichani et al., 2024) and has been very fruitful for understanding how these models work internally.
>
> **Limited novelty.** Existing results (Ekbote et al., 2025; Rajaraman et al., 2024) establish that transformers *can count* n-gram occurrences (expressivity). We address a different question: what do transformers *actually do* in the finite-sample regime where exact counts are sparse? The answer, smoothing via interpolation across partial context matches, not pure counting, is the novel contribution. More broadly, induction heads are among the most important computational primitives identified in transformers (Elhage et al., 2021; Olsson et al., 2022), yet prior work has characterized them primarily as a *copying* mechanism — matching and retrieving tokens from context. Our work reinterprets induction heads as *statistical estimators* that implement principled smoothing, connecting them to a 40-year tradition in statistical NLP (Jelinek & Mercer, 1980; Katz, 1987; Chen & Goodman, 1999). This shift from "copying circuit" to "smoothing estimator" is, to our knowledge, a novel perspective that substantially deepens our understanding of what these ubiquitous primitives compute.
>
> **Experiments reproduce theory.** We have significantly expanded the experimental section beyond the theoretical construction. First, we added new experiments with standard transformers that share no architectural specificity with the disentangled model — there is no closed-form prediction for what these models should learn, yet they recover the same mechanism, constituting independent evidence. Second, we expanded the hierarchical Dirichlet experiments from a single sequence length (original Figure 35) to all 9 sequence lengths (32–1024) with multiple seeds, providing a much more comprehensive evaluation. Third, our theoretical contribution is itself more general than pure expressivity: we characterize the *functional form* of the estimator (Proposition 3.1), connect it to classical smoothing (Section 4), and prove that gradient flow under label symmetry constrains learning to this family (Section 5). Validation in large pretrained language models on text data is an exciting direction but is beyond the scope of this theoretical work, as is standard in the ICL theory literature.
>
> **Writing transitions.** We will improve the narrative flow better connecting the different sections.
>
> **Key question: architecture dependence.** Our theory requires the disentangled architecture for analytical tractability. However, the new standard transformer experiments demonstrate that the same mechanism — both in KL performance (supplement, Figure 1) and in the internal attention patterns (Layer 1 copy heads, Figure 4; Layer 2 partial-match interpolation, Figures 3 and 5) — emerges in standard architectures. These experiments confirm that the smoothing mechanism is a property of the attention computation itself, not an artifact of the disentangled setup.

---

> > ### Author Rebuttal · Reviewer_PJrz · 2026-04-04
> >
> > I would like to very warmly thank the authors for having provided this very thorough answer containing many novel experiments. The only point of my review that has not been addressed is the fact that the experiments are still always conducted on order-2 Markov chains with vocabulary size n = 5 (unless I missed something). It seems this is not very realistic and also seems it should be quick to perform some experiments to test if the results still hold when one increases these parameters. I nonetheless increase my review to accept since the authors have addressed everything but this one point.

---

> > > ### Author Response · Authors · 2026-04-08
> > >
> > > Thank you for the careful and constructive review, and especially for recognizing the core contribution of the paper. We are very encouraged that the rebuttal addressed your main concerns, and in the revision we will incorporate the new standard-transformer experiments, clarify more explicitly both the broader relevance and the current experimental limitations, and address the remaining point about the restricted (k=2), (n=5) setting: these values were chosen for clarity of visualization and mechanistic analysis rather than because the mechanism is specific to that regime, and we agree that expanding the experiments to larger orders and vocabularies would further strengthen the paper.

---

### Official Review · Reviewer_e5ZJ · 2026-03-13

**Soundness:** 1
**Presentation:** 1
**Significance:** 3
**Originality:** 2
**Overall Recommendation:** 3
**Confidence:** 4

**Summary:**

The paper studies the in-context learning in Transformers on order-k Markov chains.
Prior analyses often focus on the infinite-temperature limit, where attention collapses to hard selection and the induced estimator becomes simple n-gram counting.
This paper instead studies the more realistic finitie-sample, finite-temperature regime and asks what kind of estimator a trained transformer implements there.
The paper claims that Transformers realize smoothing through 2 mechanisms: a BOS token induces additive pseudo-counts, and finite attention temperature induces interpolation across partial context matches.
The theory gives an explicit construction showing that a disentangled transformer can implement this estimator and compare this with classical estimators.
The paper also argues empirically that trained disentangled transformers behave similarly.

**Compliance With Llm Reviewing Policy:**

Affirmed.

**Final Justification:**

As noted in my response to the authors’ rebuttal, although my concerns regarding soundness have been resolved, I raised my score from 2 to 3 because the paper still requires substantial improvement in presentation.

**Key Questions For Authors:**

My main questions are already reflected in the weaknesses above. I list a few additional questions here.

Q1. In Figure 2, why is the gap between the Bayes-optimal estimator and the MLE estimator still quite large even for long sequence lengths? I thought that, as the sequence becomes long, the effect of the Dirichlet prior should become much smaller (and become negligible when the sequence length reaches 1000).

Q2. In settings where the prior over transition distributions is carefully designed, could interpolation over partial matches actually hurt prediction rather than help it?

**Limitations:**

The paper does not clearly discuss its limitations.
 The most immediate limitation is that both the analysis and the empirical validation rely on disentangled transformers.

**Strengths And Weaknesses:**

## Strength

- The paper studies an interesting question.
- The paper provides a solid proof for the thoery.



## Weakness

My main concerns are about soundness and presentation.

### W1. Soundness

- The formal result in Section 3 is an **expressivity** result; it shows that there exists a two-layer disentangled Transformer that realizes Equation 8. However, the paper’s main claim is stronger: it suggests that trained transformers actually implement smoothing in this way. To support this strong claim, the paper needs much stronger empirical evidence than it currently provides. In the main page, the evidence relies mostly on KL-divergence plots in Figure 2. In my view, showing that trained models achieve KL values close to the Bayes-optimal estimator is not enough, because it does not show that the model is using the same mechanism proposed in theory. A different mechanism could still produce similar KL, and thus I thought the claim of the paper less persuasive.
- In contrast, the appendix visualizations are more informative, especially Section B.3 and Section B.4 (Figures 26 and 34). These figures give more direct evidence about how the trained model is actually working. However, when I looked at them closely, I felt that they were not fully aligned with the paper’s main narrative. In Figure 34 (no BOS), the trained model places noticeable attention on partial matches, which is consistent with interpolation-based smoothing. In Figure 26 (BOS), the attention appears much sharper and is concentrated on exact matches. Based on these results, a more natural conclusion seems to be the following: **Without BOS, the model smooths mainly by interpolating across partial context matches; With BOS, it provides a
dedicated attention sink, allowing heads to attend to a neutral position when no relevant context is available, so it can rely much more on sharp exact-context matching.** This is not quite how the current paper presents the result. In particular, Proposition 3.1 keeps interpolation terms even when BOS is present.
- Also, all of the theoretical and empirical results rely on disentangled transformers. I think using disentangled transformers for theory is reasonable because the simplified architecture makes the analysis tractable. However, on the empirical side, I think the paper could do more. At least some validation on standard transformers seems feasible.

### W2. Presentation

- The paper does not discuss prior work clearly enough. On the specific topic of in-context learning of transformers on Markov chains, there is already substantial related literature, including [1, 2, 3], but the discussion in the paper appears limited. The paper also does not clearly discuss prior work on disentangled transformers.
- I also found several presentation issues that make the paper feel under-polished. First, I do not understand why the hierarchical Dirichlet prior is introduced in the main text. Its empirical role is very limited, and the corresponding result only appears in Figure 35 in the appendix. The caption of that figure even says that the figure contains a typo. It is also unclear why that figure seems to show only one model point, unlike Figure 2, where multiple model points are shown. In addition, the paper states at the beginning of Section 3 that it studies “a two-layer disentangled transformer with a single attention head,” but that statement is not accurate. Overall, I do not think the paper is polished enough for acceptance in its current form.

---
[1] Nichani, Eshaan, Alex Damian, and Jason D. Lee. "How transformers learn causal structure with gradient descent." ICML 2024

[2] Ekbote, Chanakya, et al. "What One Cannot, Two Can: Two-Layer Transformers Provably Represent Induction Heads on Any-Order Markov Chains." NeurIPS 2025

[3] Rajaraman, Nived, et al. "Transformers on markov data: Constant depth suffices." NeurIPS 2024

---

> ### Author Rebuttal · Authors · 2026-03-31
>
> We thank the reviewer for the detailed feedback. We provide supplementary material with all new experiments at this [link](https://anonymous.4open.science/r/icml_2026_rebuttal_supplementary-BCD5/main.pdf). We address each concern below; we believe the original text may not have conveyed our evidence clearly enough, and we hope the clarifications below resolve these concerns.
>
> **W1 Soundness: expressivity vs learned mechanism.** We agree that KL performance alone is insufficient to establish that the model uses a specific mechanism, and this is precisely why we included an extensive mechanistic analysis in Appendix B. to directly compare the attention mechanism of trained transformers with our construction. These visualizations confirm that the trained models form representations matching the theory, modulo invariant weight transformations. We acknowledge that this evidence was insufficiently highlighted in the main text, and will add a paragraph to the experiments section summarizing these direct mechanistic findings.
>
> **W1 Standard transformers (new experiments):** We have run new experiments with standard transformers, removing all architectural specificities of the disentangled construction. These models achieve the same KL performance as the disentangled transformer on the hierarchical Dirichlet task, outperforming all add-$\alpha$ baselines. Moreover, mechanistic analysis reveals that the standard transformer learns the same mechanism: Layer 1 heads develop sharp copy patterns at specific lags, and the Layer 2 head implements context matching with non-negligible attention on partial matches (supp., Fig. 1 for KL comparison; Fig. 3 for induction head analysis; Figs. 4–5 for attention patterns). This provides direct evidence that the interpolation behavior is not an artifact of the disentangled architecture, but rather a general property of the attention computation.
>
> **W1 BOS interpretation.** We thank the reviewer for this observation. The contrast between Figs. 26 (BOS) and 34 (no BOS) is indeed the right comparison to make, and the reviewer's reading is, in fact, exactly the prediction of our theory. We agree that the original text did not make this sufficiently clear, which may have led to the impression of a discrepancy. To clarify: Corollary 4.1 shows that the BOS token provides smoothing via pseudo-counts. When BOS is available, the model no longer needs to rely on interpolation across partial context matches for regularization, so it learns large $\beta$ values that concentrate attention on exact matches — precisely the sharp pattern observed in Fig. 26. Without BOS (Fig. 34), this pseudo-count pathway is absent, and the model must instead smooth through partial context interpolation at finite $\beta$, producing the broader attention pattern the reviewer noted. Proposition 3.1 retains the interpolation terms in both cases for generality, but the key insight, which we will state explicitly in the revised text, is that BOS and partial-match interpolation are *complementary* smoothing mechanisms. We will revise the relevant passages to make this interpretation unambiguous and to highlight that Figs. 26 and 34 together provide evidence for the theory.
>
> **W2 Related work.** We apologize for the oversight. We will add a detailed discussion of the related works, as well as mentions in the introduction, to better place our contributions in the context of previous works.
>
> **W2 Hierarchical Dirichlet (new experiments).** We agree that the hierarchical prior was under-supported empirically. The hierarchical Dirichlet is crucial because it is the setting where add-constant smoothing is not Bayes-optimal, making the benefit of interpolation visible. In the revised paper, we will (1) move the hierarchical results from the appendix into the main text and (2) expand the experiments: the original Fig. 35 showed results for only one sequence length; the new version (supp., Fig. 1) includes all sequence lengths. We have also (3) added new standard transformer results on this task (supp., Fig. 1). The caption typo will be fixed.
>
> **W2 "Single head" statement.** Thank you, we will fix the typo.
>
> **Q1 KL gap.** With vocabulary size $|V|=5$ and order $k=2$, there are $5^2 = 25$ possible contexts. At sequence length 1024, each context appears approximately 40 times on average. The gap decreases monotonically with sequence length, but does not become negligible until sequences are substantially longer.
>
> **Q2 Can interpolation hurt?** By definition, the interpolation estimator is not Bayes-optimal in general, so there exist priors under which it is strictly suboptimal. However, when compared to the MLE, one would need to design rather exotic priors for interpolation to perform worse: any form of smoothing prevents the KL divergence from diverging when the MLE assigns zero probability to an observed transition, which is the dominant failure mode of count-based estimators in the finite-sample regime.

---

> > ### Author Rebuttal · Reviewer_e5ZJ · 2026-04-04
> >
> > I thank the authors for the rebuttal. After re-reading the paper and the authors’ response, I agree that the paper does describe these two mechanisms as providing complementary forms of smoothing. My main concern regarding soundness has been resolved, and I have raised my score from 2 to 3.
> >
> > The reason I did not raise the score further to 4 is that the paper requires substantial improvement in presentation. As the authors themselves acknowledged in their response,
> >
> > - the attention visualizations in Appendix B should be moved, at least in part, into the main text;
> > - the BOS interpretation should be presented more clearly;
> > - the hierarchical Dirichlet prior results should be strengthened;
> > - the related work section requires substantial improvement.

---

> > > ### Author Response · Authors · 2026-04-08
> > >
> > > Thank you again for taking the time to reread the paper and our rebuttal. We appreciate that you found the main soundness concern resolved and updated your score accordingly.
> > >
> > > We understand your remaining concerns about presentation, and we agree that the paper would benefit from clearer exposition in several places. In particular, we agree that the attention visualizations in Appendix B should be given more prominence in the main text, that the BOS interpretation should be explained more explicitly, and that the hierarchical Dirichlet results should be presented more clearly and comprehensively. These are all points we are committed to addressing in the revision.
> > >
> > > At the same time, we hope the rebuttal helped clarify that the central claims of the paper are supported. In particular, we clarified the mechanistic interpretation, explained the BOS/no-BOS contrast, and added new experiments on standard transformers and on the hierarchical Dirichlet setting. We fully agree that these points should be communicated more clearly in the paper, and we will make those improvements in revision.
> > >
> > > On the hierarchical Dirichlet point in particular, we agree that this aspect was under-emphasized in the original submission. This is precisely why we added expanded results in the supplementary material, including multiple sequence lengths and standard-transformer comparisons. We agree that these results should be incorporated more prominently into the paper.
> > >
> > > We are grateful for your feedback, and we hope you might take into account that these remaining concerns are ones we can address directly in a revision.

---

### Official Review · Reviewer_K9ZP · 2026-03-13

**Soundness:** 4
**Presentation:** 3
**Significance:** 4
**Originality:** 3
**Overall Recommendation:** 5
**Confidence:** 3

**Summary:**

This work examines the kind of in-context estimator transformers implement through the lens of order-k markov chains and disentangled transformers, linking transformers' in-context learning capabilities to classical statistical smoothing. The paper walks through a two-layer transformer by construction that solves order-k markov chain prediction through extended induction-head-like mechanisms: prior-context concatenation in the first layer and context-matching (interpolation between multiple levels of partial match) in the second layer. Empirically-trained disentangled transformer behaviorally match the constructed one as well as the bayes-optimal prediction. The paper additionally elucidates the role of the BOS token, and extends beyond markov-chain sequences.

**Compliance With Llm Reviewing Policy:**

Affirmed.

**Final Justification:**

The rebuttal reaffirmed my positive recommendation.

**Key Questions For Authors:**

- What happens when you train a standard transformer on the same order-k markov chain tasks? Would you expect any differences, e.g. when there is ample redundancy in head capacity or representational capacity?
- Have you considered testing this in the label symmetry setting?
- Can this approach be extended to account for tensions and tradeoffs between in-context and in-weight learning?
- What are the key implications for modern LLMs? Would it be possible to test if signatures of this kind of context matching is happening in pre-trained language models?

**Limitations:**

The paper currently lacks a discussion of its limitations.

**Strengths And Weaknesses:**

Strength:
- This work provides significant insights in understanding the mechanisms of in-context learning in transformers.
- The investigation is thorough -- the paper walks through the proof by construction in detail, then presents strong empirical behavioral and mechanistic evidence, as well as considering more general data distributions beyond the order-k markov chain.
- The presentation is dense but overall clear.

Weakness:
- The paper lacks discussions of the broader implications for modern LLMs.

---

> ### Author Rebuttal · Authors · 2026-03-31
>
> We thank the reviewer for the positive assessment and thoughtful questions. We provide supplementary material with all new experiments at [this anonymous link](https://anonymous.4open.science/r/icml_2026_rebuttal_supplementary-BCD5/main.pdf).
>
> **Standard transformers (new experiments)(Q1).**
>
> Whether additional representational capacity changes the learned estimator depends on the task. Under the independent Dirichlet prior, the Bayes-optimal predictor is the add-$\alpha$ estimator, which the disentangled transformer already implements exactly (Corollary 4.1). Since optimality is already achieved, additional capacity cannot improve performance — it can only find alternative parameterizations of the same estimator. The hierarchical Dirichlet setting is more revealing: because the Bayes-optimal predictor has no closed-form expression, increased capacity could in principle discover a better approximation to the true posterior predictive distribution.
>
> To test this empirically, we have added experiments with **standard transformers** (learned token embeddings with $d_{\text{model}}=64$, additive residual connections, LayerNorm, MLP layers, and learned relative positional biases) trained on the hierarchical Dirichlet task — removing all architectural specificities of the disentangled construction. These models achieve the same KL performance as both the disentangled construction and the fully trained disentangled transformer, substantially outperforming all add-$\alpha$ baselines (supplement, Figure 1 for KL comparison; Figures 3–5 for mechanistic evidence). Crucially, mechanistic analysis confirms the same two-stage mechanism: Layer 1 heads learn sharp copy patterns at specific lags, and the Layer 2 head implements context matching with non-trivial attention on partial matches — the same interpolation behavior predicted by our theory. This demonstrates that the mechanism is a general property of the attention computation, not an artifact of the disentangled architecture.
>
> **Label symmetry (Q2).** Section 5 and Corollary 5.2 already provide a theoretical generalization: under any data distribution with label symmetry, gradient flow constrains the two-layer disentangled transformer to learn estimators of the same soft context-matching, but with generalized masks $M \subseteq [k] \times [k]$.
>
> Could the reviewer clarify the question further?
>
> **In-context vs in-weight learning (Q3).** In our setting, each sequence is generated from a freshly sampled Markov chain, so the model must perform pure in-context learning. However, the learned temperature $\beta$ and BOS parameters can be viewed as in-weight knowledge about the prior structure (e.g., the Dirichlet concentration). Extending the analysis to settings where the model also memorizes specific transition matrices, and characterizing the resulting tension, is an interesting direction for future work.
>
> **LLM implications (Q4).** Induction heads are known computational primitives in large language models (Olsson et al., 2022). Our smoothing interpretation suggests that LLMs may implement similar interpolation over partial context matches when predicting in repetitive or structured text. Concretely, one could probe attention patterns in pretrained models on text with known repetitive structure and test whether attention weights correlate with partial context overlap as our theory predicts. We will add a discussion of broader implications and limitations to the revised paper.
>
> **Limitations.** We will add a dedicated limitations paragraph discussing the restriction to synthetic Markov chain data and highlight the theoretical nature of the work. We note that validating on large models with text data, while exciting, is beyond the scope of this theoretical contribution and is left for future work.

---

> > ### Author Rebuttal · Reviewer_K9ZP · 2026-04-03
> >
> > I thank the authors for following up to these questions. I will keep my accept recommendation.

---

> > > ### Author Response · Authors · 2026-04-08
> > >
> > > Thank you for the very positive assessment and for the thoughtful questions. We are especially encouraged that you found both the theoretical construction and the empirical/mechanistic evidence convincing, and in the revision, we will make sure to strengthen the discussion of limitations and broader implications for standard transformers and modern LLMs.

---

### Decision · Program_Chairs · 2026-04-30

**Decision:**

Accept (regular)

**Comment:**

The paper proves that two-layer disentangled transformers trained on order-$k$ Markov chains implement a soft context-matching estimator interpolating over partial matches. Beginning-of-sequence (BOS) tokens act like Dirichlet pseudo-counts, and finite attention temperature gives Jelinek--Mercer-style smoothing. This reframes induction heads as smoothing estimators rather than just copying circuits.

Three reviewers gave accept (5).

Reviewer e5ZJ was mainly concerned about the reliance on the disentangled architecture and presentation. The rebuttal handled the first concern well with new standard-transformer experiments showing the same mechanism emerges without the disentangled setup.

However, I agree with the Reviewer e5ZJ  that the paper needs improvement on the presentation side.  The BOS interpretation should be stated more clearly, and the related work section needs substantial expansion. I also noticed that many equations need minor polishing (missing commas or periods after equations). Please work on these in the revision.